# Tackling Unconditional Generation for Highly Multimodal Distributions with Hat Diffusion EBM

## Abstract

This work introduces the Hat Diffusion Energy-Based Model (HDEBM), a hybrid of EBMs and diffusion models that can perform high-quality unconditional generation for multimodal image distributions. Our method is motivated by the observation that a partial forward and reverse diffusion defines an MCMC process whose steady-state is the data distribution when the diffusion is perfectly trained. The components of HDEBM are a generator network that proposes initial model samples, a truncated diffusion model that adds and removes noise to generator samples as an approximate MCMC step that pushes towards realistic images, and an energy network that further refines diffusion outputs with Langevin MCMC. All networks are incorporated into a single unnormalized density. MCMC with the energy network is crucial for driving multimodal generation, while the truncated diffusion can generate fine details needed for high-quality images. Experiments show HDEBM is effective for unconditional generation with sampling costs significantly lower than diffusion models. We achieve an FID score of 21.82 on unconditional ImageNet at 128x128 resolution, which to our knowledge is state-of-the-art among unconditional models which do not use separate retrieval data.

## 1 Introduction

Image generation with deep learning has made tremendous progress in the past decade as models become more sophisticated and available compute increases. Conditional modeling, where auxiliary information such as a label or text is used to guide model synthesis, has led to impressive results for applications such as class-conditioned [3, 7] and text-conditioned [44, 45] generation. While unconditional modeling has also seen great progress, unconditional models often significantly underperform conditional counterparts. This is especially true for highly multimodal[1] and high-resolution datasets such as ImageNet [6]. Improvements in unconditional modeling techniques have the potential to increase our understanding of non-convex probability densities, enable better generation when supervised information is unavailable, and increase the diversity of conditional generations.

Generative adversarial networks (GANs) [13] and diffusion models [21] are the most popular current methods for high-resolution image synthesis. Both have drawbacks when it comes to highly multimodal unconditional modeling. The adversarial objective of GAN learning often leads to the mode collapse phenomenon [34] where the generator model only learns to generate a small subset of the entire data distribution. Diffusion models face the challenge of assigning samples to separate modes early in the reverse diffusion process when image features first start to emerge from noise. The initial phase of the reverse diffusion process can be very sensitive to network changes. Using an exponential moving average (EMA) of the weights can help alleviate this sensitivity by stabilizing outputs across model updates and it is typically a vital part of diffusion learning.

---

[1]In this work multimodal refers to diversity of images in a dataset rather than separate data domains.

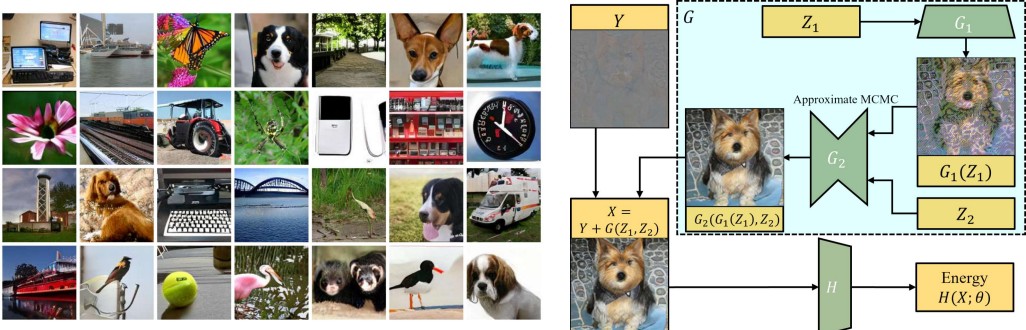

Figure 1: *Left:* Unconditional ImageNet 128x128 samples generated by HDEBM. *Right:* Visualization of energy function structure in HDEBM. $G_1$ creates an intial image from noise $Z_1$ which is passed through a forward/reverse truncated diffusion in $G_2$. $G_2$ then performs approximate MCMC on the data distribution. The output is adjusted with residual image $Y$ to create a visible image $X$ for forward pass energy calculation with $H$. Sampling uses Langevin MCMC via backpropagation.

We explore the potential of the energy-based model (EBM) as a method for highly multimodal unconditional learning. While existing EBMs often do not match the performance of GANs and diffusion models for low-resolution data, the recently introduced Hat EBM [19] showed surprisingly strong performance on high-resolution unconditional ImageNet generation. Nonetheless, Hat EBM does not achieve state-of-the-art results. In this work, we build upon Hat EBM to develop a new model that achieves state-of-the-art unconditional synthesis for ImageNet 128x128.

A fundamental obstacle of EBM learning is the computational burden of the MCMC inner loop used for each model update. Computational restrictions allow only shortrun MCMC trajectories during training, limiting the fine image details the EBM generates. Diffusion models decouple learning from MCMC sampling and thereby can use many steps during test-time generation to create fine image details. Incorporating diffusion sampling into the EBM sampling process has the potential to greatly improve generation quality while preserving the relatively fast sampling speed and wide mode coverage of EBMs.

Our key insight is that adding noise to an image and removing noise with a perfectly trained diffusion model defines an MCMC trajectory whose steady-state is the data distribution (see Section 3.2). Building on this, we propose to add and remove noise from generator samples using a truncated diffusion model as an initial approximate MCMC step, followed by further Langevin MCMC refinement from the EBM. To enable the truncated diffusion to be incorporated into EBM learning, we train it separately and distill it to a single step using progressive distillation [48]. Similar to the approach in [60], we only train the truncated part of the diffusion model near the data distribution and ignore the higher noise levels. This bypasses the most challenging parts of diffusion learning and greatly reduces the size of the diffusion model without sacrificing denoising performance. The difficulty of the training and distillation process is greatly reduced for truncated diffusion compared to full diffusion. Once trained, the truncated and distilled diffusion is incorporated into Hat EBM between the generator and energy network. It can be viewed both as an extension of the generator that refines the base generator output and as an extension of the MCMC sampling phase using an approximate sampling step. We call this model the Hat Diffusion EBM (HDEBM). Experiments show HDEBM significantly improves sample quality compared to Hat EBM without significant increase in computation cost beyond training the truncated diffusion. Curated samples from HDEBM along with model energy structure are shown in Figure 1. In summary, our main contributions are listed below.

- We introduce the novel perspective that a partial forward and reverse process for a perfectly trained diffusion model defines an MCMC trajectory whose steady-state is the data distribution.

- We develop the Hat Diffusion EBM (HDEBM) modeling framework that incorporates a truncated and distilled diffusion into Hat EBM to help train the generator and energy network. All networks are incorporated into a single unnormalized density.

- Experiments on CIFAR-10 [24] and Celeb-A 64x64 [30] show that HDEBM has state-of-the-art synthesis performance among explicit EBMs. Experiments on unconditional ImageNet 128x128 show that, to our knowledge, Hat EBM has state-of-the-art synthesis performance among models that do not use retrieval data during test time.

## 2 Related Work

**EBMs.** Energy Based Models (EBMs) are a class of generative models that aim to represent a data distribution with an unnormalized density. Early work includes Boltzmann machines [1, 46], and the Filters Random field And Maximum Entropy (FRAME) model [61]. Recent advancements in deep learning have led to investigations in using Convolutional Network based EBM models [58] increasing the image synthesis [38, 10] abilities. The community has also trained EBMs with auxiliary models. One approach trains the EBM using direct outputs from the generator without MCMC sampling [23], which is further explored by methods such as EGAN [5] and the VERA model [14]. Cooperative learning [55] uses a generator to initialize MCMC samples used to train the EBM and uses a reconstruction loss between the EBM samples and the generator output to update the generator. EBMs defined in latent space [39, 40] have also been explored as the energy landscape in latent space can provide better movement along the complex image manifold. The closest work to our approach is Hat EBM [19] which builds upon [55] to incorporate the generator latent space into the unnormalized density. We provide a comparison between Hat EBM and HDEBM in Appendix C.1.

**Diffusion Models.** Diffusion models [49, 21, 50] are based on the notion of adding and removing noise in order to learn underlying patterns of a dataset. The slow sampling speed of early models has been significantly expedited with acceleration techniques [50, 48, 59, 31, 60, 32], several of which are related to our approach. DDIM [50] employs a class of non-Markovian diffusion processes to define a faster deterministic sampling method. Truncated diffusion trajectories, wherein only fragments of the forward and reverse processes are performed, have been appended to other kinds of generative models to improve sample quality [60, 32]. Truncated diffusion models have also found applications in image editing [33] and adversarial defense [37]. We build upon these works by noting that an ideal truncated diffusion defines an approximate MCMC process with the data distribution as its steady-state, which can serve as a tool for instructing other networks. A comparison between HDEBM and other methods [60, 32] that employ truncated diffusion for generation is provided in Appendix C.2. Progressive distillation [48] trains a series of student networks to match the DDIM paths of teacher networks. The distilled model obtains high quality samples with only a few steps.

## 3 Method

This section first presents background theory for learning EBMs and for learning and distilling diffusion models. Next, we discuss how truncated and distilled diffusion can be viewed as an efficient approximate MCMC update. Finally, HDEBM model formulation and training methods are presented.

### 3.1 Background

**EBM Learning.** Our EBM learning follows the methods from [20, 57, 61]. An EBM is defined as

$$p(x; \theta) = \frac{1}{\mathcal{Z}(\theta)} \exp\{-U(x; \theta)\} \tag{1}$$

where $U(x; \theta)$ is a deep neural network with parameters $\theta$, $x$ is an image state, and $Z(\theta)$ is an intractable normalizing function. A maximum likelihood objective is used to minimize the Kullback–Leibler divergence $\mathrm{argmin}_\theta D_{KL}(q_0(x) \parallel p(x; \theta))$, where $q_0(x)$ is the true and unknown data distribution, by using stochastic gradient descent

$$\nabla\mathcal{L}(\theta) \approx \frac{1}{n}\sum_{i=1}^{n}\nabla_\theta U(X_i^+; \theta) - \frac{1}{n}\sum_{i=1}^{n}\nabla_\theta U(X_i^-; \theta) \tag{2}$$

where $X_i^+$ are samples from the data distribution and $X_i^-$ are samples from the model $p(x; \theta)$. To obtain samples from the model, MCMC sampling with $K$ steps of the Langevin equation is used:

$$X^{(k+1)} = X^k - \frac{\epsilon^2}{2}\nabla_{X^{(k)}}U(X^{(k)}; \theta) + \epsilon V_k \tag{3}$$

where $X^{(k)}$ is the sample at step $k$, $\epsilon$ is the step size, and $V_k \sim N(0, I)$. Generating negative samples also requires an initialization strategy to obtain the initial states $\{X_{i,0}^-\}_{i=1}^n$.

**Diffusion Learning and Distillation.** This section provides a concise review of diffusion models, truncated diffusion, and distilled diffusion. We denote data distribution as $X \sim q_0$ and consider $q_t$ for $t \in [0, T]$ as the forward process which produces noisy samples by adding Gaussian noise to

the data. Specifically, the noisy samples $x_t$ can be parameterized given $\alpha_t$ and $\sigma_t$, such that the log signal-to-noise-ratio, $\lambda_t = \log(\alpha_t^2/\sigma_t^2)$, decreases monotonically over time $t$. The forward process can be defined by a Gaussian process constituting a Markov chain:

$$q_t(x_t|x) = N(x_t; \alpha_t x, \sigma_t^2 I), \quad q_t(x_t|x_s) = N(x_t; (\alpha_t/\alpha_s)x_s, \sigma_{t|s}^2 I), \tag{4}$$

where $0 \leq s < t \leq T$ and $\sigma_{t|s}^2 = (1 - e^{\lambda_t - \lambda_s})\sigma_t^2$. To sample from data distribution $q_0$, we first sample from $q_T$ then sample reverse steps until we reach $x$. As suggested by [21] and following works, we can construct a neural denoiser $\hat{x}_\theta(x_t)$ to predict an estimate of $x$, and learn a model using a weighted mean-squared-error loss:

$$\mathcal{L}(\theta) = E_{X \sim q_0, t \sim U[0,T], x_t \sim X_t(\cdot|x)}[w(\lambda_t)\|\hat{x}_\theta(X_t) - X\|_2^2]. \tag{5}$$

In this work, we train truncated diffusions which only use part of the forward/reverse process as in [60]. This simply involves changing the sampling distribution of $t$ from $U[0,T]$ to $U[0,T']$ for $T' < T$. We limit our training to either the final $T' = 256$ or $T' = 512$ steps of a discrete cosine schedule with $T = 1000$ steps.

A DDIM sampler [50] can achieve fast, high-quality, and deterministic generation. Our works utilizes progressive distillation of DDIM to further accelerate sampling [48]. Student models are trained so that one DDIM step of the student model matches two DDIM steps of the teacher model. The initial teacher is the full diffusion. After convergence the student becomes the new teacher, halving the number of steps until the entire denoiser consists of a single step.

## 3.2 Truncated Diffusion as MCMC Sampling

In this section we develop a theoretical understanding of the truncated diffusion process used as part of the HDEBM. We begin with a straightforward proposition that demonstrates the central claim.

**Proposition 3.2.1.** *Suppose $D(x,t)$ is the DDIM reverse process starting at timestep $t$ for a perfectly trained diffusion model, meaning that if $X' \sim q_t$ then $D(X',t) \sim q_0$. Further suppose that the support of $q_0$ is contained within a compact set. Then the stochastic update*

$$X \mapsto D(\alpha_t X + \sigma_t Z, t) \quad where \quad Z \sim N(0,I) \tag{6}$$

*defines an ergodic MCMC process with a unique steady-state distribution $q_0$ for any timestep $t$.*

*Proof.* It is clear that (6) is a Markov transition since the updated state only depends on the starting state $X$ and noise. The forward process by definition has the property that $X' = \alpha_t X + \sigma_t Z \sim q_t$ if $X \sim q_0$ and $Z \sim N(0,I)$. Therefore if $X \sim q_0$ then $Y = D(\alpha_t X + \sigma_t Z, t) \sim q_0$ under the assumption that $D$ is perfectly trained. This shows that the Markov update (6) preserves the data distribution $q_0$. We note the process is aperiodic and irreducible because adding Gaussian noise to $X$ can map to any image state with non-zero probability, and the assumption that $D$ is perfect means there is always some image in $q_t$ that will map to a given image in the support of $q_0$. Since the support of $q_0$ is contained in a compact set, the chain must be recurrent so that the process is ergodic and $q_0$ is the unique steady-state. $\square$

We view the proposition as a useful insight that allows a more principled framework for using truncated diffusion than the empirical perspective presented in prior works. SDEdit [33] empirically observes that a truncated diffusion process can add realism to naive edits or rough user-defined templates. Understanding the truncated diffusion as an approximate MCMC step on the data distribution gives a clearer picture of why this occurs since we expect MCMC to move out-of-distribution states towards the data distribution while still retaining some features of the original state due to MCMC autocorrelation. The same observation applies to the DiffPure defense [37] which uses truncated diffusion to remove adversarial signals while preserving most of the original image appearance.

The proposition applies to any timestep $t > 0$. The value of the $t$ determines how far the MCMC step travels across the data distribution. In the limiting case $t = T$ the MCMC step samples independent images from the full diffusion each time. Large values of $t$ greatly changes image appearance in each step (lower MCMC autocorrelation) while small values of $t$ retains more of the original image appearance (higher MCMC autocorrelation). This is analogous to the discussion of the role of noise in SDEdit [33]. Additionally, we expect that truncated diffusions are much easier to learn when $t$ is small and much harder to learn when $t$ is large because smaller $t$ define an easier denoising problem while larger $t$ require more coordination between timesteps to drive noisy samples to different modes.

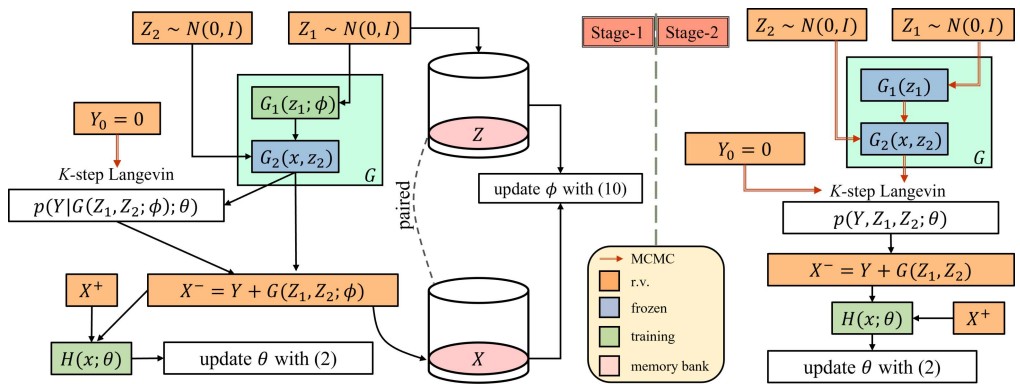

Figure 2: Visualization of Stage 1 (left) and Stage 2 (right) training methods for HDEBM. Red arrows indicate that the initial random variable is updated using Langevin MCMC according to the given density. Langevin with $p(Y|G(Z_1, Z_2; \phi); \theta)$ and $p(Y, Z_1, Z_2; \theta)$ uses the same equations as [19].

We further develop truncated diffusion as a modeling tool by observing a synergy between learning a truncated diffusion and learning a distilled diffusion. As noted in [48], a challenging aspect of learning a distilled diffusion is that the diffusion network output for a noise image at $t = T$ provides essentially no information about the final state before distillation, while once the model is distilled to a single step the final image must be fully predicted from noise. When distilling a truncated diffusion, noisy images at $t = T'$ have many features of the original image and diffusion network outputs can retain many of these features throughout distillation while refining overall appearance. In contrast to full diffusion distillation [48] we find only a minor performance gap between the undistilled truncated diffusion and the truncated diffusion distilled to a single step.

In summary, we view truncated and distilled diffusion as an efficient tool that can perform approximate MCMC sampling with updates that can travel much further along with image manifold than conventional methods like Langevin MCMC. After distillation, it becomes computationally feasible to perform forward/backward passes through the MCMC step to train other networks. There is a significant need for MCMC tools with better movement along complex manifolds and truncated diffusion MCMC has the potential to fill this gap. We note there are several challenges before truncated diffusion MCMC can be a general purpose tool. The process is approximate in practice, non-reversible, and lacks an explicit transition density function so that Metropolis-Hastings correction is not immediately applicable. In this work, we show that the unadjusted process is a useful tool for teaching generator and energy networks. We hope that this tool can be adapted into a rigorous and general purpose MCMC transition in future works.

### 3.3 Hat Diffusion EBM

This section describes the HDEBM model formulation and training process. We assume that a truncated and distilled diffusion network $D(x)$ that approximately maps the noisy distribution $q_{T'}$ to the data distribution $q_0$ in a single forward pass has been trained and frozen. The choice of $T'$ depends on the dataset. We first adapt the synthesis-oriented training of Hat EBM to jointly train the energy network and generator network. In this stage of training, samples are generated by drawing latent Gaussian random variables and performing MCMC on a residual image conditioned on frozen latents. We then perform a second stage of training that adapts the energy function from a form where latents must be frozen after initialization to a form where both latents and image residuals can be updated with MCMC sampling. The ability to perform MCMC refinement in both the image and latent space can greatly improve sample quality over the first stage model. The second stage model has more appealing properties as an explicit energy model since the intractable normalizer does not depend on the latent state.

Following the Hat EBM formulation, we make the assumption that data samples $X \sim q_0$ can be decomposed as $X = G(Z) + Y$ where $G$ is a generator network, $Z$ is a latent random variable, and $Y$ is a random variable which functions as a residual image to bridge the gap between the generator output manifold and the data manifold. In the first stage of learning we assume $Z \sim N(0, I)$ and we learn the distribution $Y|Z$. In the second stage we learn the joint distribution of $Z$ and $Y$.

205   Our central modification is to define $G$ as

$$G(z) = G_2(G_1(z_1), z_2), \quad \text{with} \quad G_2(x, z) = D(\alpha_{T'} x + \sigma_{T'} z) \tag{7}$$

206 where $z = (z_1, z_2)$. $G_1$ creates initial image proposals from noise which are refined by the forward
207 and reverse process of $G_2$. Depending on the context we use the notation $G_1(z_1)$ to show a fixed
208 generator and $G_1(z_1; \phi)$ to show a learnable generator with weights $\phi$. When $G_1$ is learnable we
209 denote the entire generator as $G(z_1, z_2; \phi)$. The noising and denoising process applied to $G_2$ can be
210 interpreted as an MCMC step that pushes the initial generator output closer to the data distribution.
211 In the second stage, we refine both $z_1$ and $z_2$ using MCMC initialized from $N(0, I)$. A diagram of
212 each training stage is show in Figure 2.

### 3.3.1   First Stage: Residual Distribution Conditioned on Fixed Latent

214 This section describes how to train a model that can create samples by drawing a Gaussian random
215 variable $Z \sim N(0, I)$, passing $Z$ through a generator network to create initial images, and then
216 refining these image samples using MCMC with an energy network while leaving latent variables
217 fixed. The methodology takes inspiration from cooperative learning [55] and Hat EBM [19]. It is
218 difficult to formulate a single maximum likelihood learning framework to train both the generators
219 and energy. Therefore we follow prior work and use two maximum likelihood objectives: one to train
220 the energy network assuming the generator is fixed and another to train the generator assuming the
221 energy is fixed. Intuitively the energy objective will teach the EBM the best way to refine a fixed
222 generator and the generator objective will teach the generator how to best close the gap between its
223 current samples and refined EBM samples. The generator is trained using purely synthetic data from
224 its own outputs and EBM refinement. Following prior work, in practice we alternate between updates
225 of the EBM and generator.

226 To train the energy function, we assume that $G_1$ and $G_2$ are both fixed and use $G(Z)$ to denote the
227 entire generator process (7). The model density is given by

$$p(y, z; \theta) = \frac{1}{\mathcal{Z}_z(\theta)} \, p_0(z) \exp\{-H(G(z) + y; \theta)\} \tag{8}$$

228 where $H(x; \theta)$ gives the energy output from the sum of the generator and residual image and $p_0$ is the
229 $N(0, I)$ density. Learning the weights $\theta$ is identical to Hat EBM learning with a different generator
230 structure. To obtain negative samples, we initialize $Z^- \sim N(0, I)$ and $Y^- = 0$ and then use shortrun
231 Langevin sampling (about 50 steps) to obtain $Y^-|Z^-$. We assume the data distribution has the form
232 $X = G(Z) + Y$ where $Z \sim N(0, I)$ and $Y|Z \sim q_0(y|z)$ for some unknown distribution. Learning
233 uses the standard EBM update (2) with an energy form $U(y|z; \theta) = H(G(z) + y; \theta)$ where data
234 samples $X^+ \sim q_0$ are sufficient statistics for updating $\theta$ and $(Y^+, Z^+)$ do not need to be inferred.

235 To update the generator $G_1(z_1; \phi)$, we assume that we have a fixed energy network $H(x)$ and
236 a fixed generator $G(z)$ from the current model. We treat the shortrun MCMC process with the
237 density (8) used to generate negative samples as the ground truth distribution. Specifically, while
238 updating the generator we assume that the data distribution is the joint distribution $(X, Z)$ where
239 $Z \sim N(0, I)$ and $X = G(Z) + Y$ where $Y$ is generated from short-run MCMC using the energy
240 $U(y|z) = H(G(z) + y)$ initialized from $Y = 0$. We denote the shortrun MCMC distribution as
241 $s(x|z)$. We aim to train the generator $G(z_1, z_2; \phi) = G_2(G_1(z_1; \phi), z_2)$ to match this distribution.
242 No real data is used to train the generator. Even with perfect learning, the samples from the updated
243 generator can be no better than samples from the current HDEBM model. The goal is instead to
244 match the current HDEBM samples with only the updated generator to provide a better MCMC
245 initialization for future HDEBM learning iterations.

246 The form of the learnable generator distribution is a key design choice of HDEBM. The latent
247 distribution is set to be $Z \sim N(0, I)$ and we learn the conditional density $p(x|z; \phi)$. We propose to
248 use two energy terms: one which encourages the output of $G(z_1, z_2; \phi)$ to be close to refined EBM
249 samples, and one which encourages the output of $G_1(z_1; \phi)$ to be close to refined EBM samples. The
250 density is given by

$$p(x|z; \phi) = \frac{1}{\mathcal{Z}} \exp\left\{-\beta_1 \|x - G(z_1, z_2; \phi)\|^2\right\} \exp\left\{-\beta_2 \|x - G_1(z_1; \phi)\|^2\right\} \tag{9}$$

251 which is the product of the Gaussians $N(G(z_1, z_2; \phi), \beta_1^{-1/2})$ and $N(G_1(z_1; \phi), \beta_2^{-1/2})$ for constants
252 $\beta_1, \beta_2$. The constants $\beta_1, \beta_2$ allow a trade-off between the importance of the energy terms. Since the

product of two Gaussians is a Gaussian whose standard deviation does not depend on the Gaussian means of the product terms, the normalizer $\mathcal{Z}$ does not depend on $\phi$ and the maximum likelihood objective can be written in closed form:

$$\phi^* = \mathrm{argmin}_\phi E_{p_0(z)s(x|z)} \left[ \beta_1 \| X - G(Z_1, Z_2; \phi) \|^2 + \beta_2 \| X - G_1(Z_1; \phi) \|^2 \right]. \qquad (10)$$

The first term will adjust the output of $G_1(z_1; \phi)$ so that the entire generator $G(z_1, z_2; \phi)$ produces a sample that resembles an EBM sample after the output of $G_1$ goes through the noising and denoising step from $G_2$. This greatly eases the burden of training the generator compared to the approach in Hat EBM. Assuming that EBM samples are close to the data distribution, the forward/reverse diffusion from $G_2$ will naturally push samples from $G_1$ towards the target distribution. $G_1$ can learn to produce any distribution whose samples are mapped to samples close to the data distribution when $G_2$ is applied. The data distribution itself is one such possibility, but there are others which are easier to learn including images which resembled smoothed data or even images with artifacts that exploit imperfections in the diffusion $G_2$. The function of this loss term can be interpreted as training $G_1$ to invert $G_2$ given forward noise $Z_2$ and target image $X$.

The second term will encourage the output of $G_1$ to match EBM without considering $G_2$. We view this term as a regularizer. Since $G_1$ can learn many possible distributions that match noisy data after forward noise is applied, this term can encourage $G_1$ to find a solution that resembles the data. In practice, we find that the first term is essential for good synthesis results while including the second term with a small $\beta_2$ can in some cases yield slight improvements. In other cases we set $\beta_2 = 0$ and only use the first term.

In practice we alternate between one update of $H(x; \theta)$ and one update of $G_1(z_1; \phi)$. Additionally, we maintain a bank of pairs $\{(X^{(i)}, Z^{(i)})\}$ and draw batches from this bank to update the generator, after which the batch states are overwritten with pairs from the most recent EBM update. As in Hat EBM, we find this is more effective than using only the most recent EBM samples to update $G_1$ because the most recent EBM samples can sometimes lack diversity while historical samples tend to have good diversity. Training the generator on samples with limited diversity can cause instability as the EBM tries to compensate by strongly adjusting the MCMC paths. We experimented with performing 10 steps to update the EBM followed by 10 steps to update the generator without the historical bank and saw good initial results. However this training strategy requires twice the amount of MCMC since fresh samples are needed to update both the generator and EBM. It also requires a second copy of the weights of $G$ in GPU memory since sampling from $s(x|z)$ requires a frozen copy if more than one generator update is used. To save memory and compute we use the historical bank approach. See Appendix D.3.1 for training pseudocode.

### 3.3.2 Second Stage: Joint Residual and Latent Distribution

The second stage of training will finetune a model $H(x; \theta)$ which is pretrained as a density of the form (8) to become a density of the form

$$p(y, z; \theta) = \frac{1}{\mathcal{Z}(\theta)} \exp\{-H(G(z) + y; \theta)\}. \qquad (11)$$

The primary difference between (11) and (8) is that the normalizer of the former depends only on $\theta$ while the normalizer of the latter depends on both $\theta$ and $z$. This means that we can perform MCMC on $z$ for the density (11) but not for (8). We use the second stage as a way to refine the initial generator appearance which might have blurs or artifacts that can be corrected by local movement. We leave both $G_1$ and $G_2$ frozen during the second stage and we initialize $\theta$ from the weights of the first phase. Negative samples are obtained from alternating Langevin steps of $Y$ and $Z$ initialized from $Y = 0$ and $Z \sim N(0, I)$. The first stage is critical for aligning the output of $G(z)$ to produce realistic images near $N(0, I)$, which provides high-quality MCMC initialization for the second phase. The EBM update uses the same equation as the first stage. We experimented with including the Gaussian prior $p_0(z)$ in (11) but found negligible effect. See Appendix D.3.2 for training pseudocode.

## 4 Experiments

We now present our HDEBM experiments for unconditional generation. All networks used in our experiments are trained from scratch using only unconditional data from a single dataset. Each experiment involves three rounds of training. First, we train the truncated diffusion and distill it to

Table 1: Comparison of FID scores among representative generative models. For CIFAR-10 and Celeb-A, all FID reports are for unconditional models. EBMs are above the dividing line and other models are below. For ImageNet, models above the dividing line are unconditional and models below use label information. (*=re-evaluated using evaluation code from [7], c.g.=classifier guidance)

| CIFAR-10 ($32 \times 32$) | | CelebA ($64 \times 64$) | | ImageNet ($128 \times 128$) | |
|---|---|---|---|---|---|
| Model | FID | Model | FID | Model | FID |
| VERA [15] | 27.5 | Divergence Triangle [16] | 31.9 | InfoMax GAN [27] | 58.9 |
| Improved CD EBM [9] | 25.1 | Hat EBM | 11.57 | Hat EBM (small) [19]* | 43.89 |
| Hat EBM [19] | 19.30 | Diff. Recov. EBM | 5.98 | Hat EBM (large) [19]* | 31.89 |
| CoopFlow [56] | 15.80 | HDEBM (Stage 1) *(ours)* | 5.55 | SS-GAN (small) [4] | 43.9 |
| VAEBM [54] | 12.19 | HDEBM (Stage 2) *(ours)* | **4.13** | SS-GAN (large) [4] | 23.4 |
| Diff. Recov. EBM [12] | 9.58 | NVAE [53] | 14.7 | HDEBM (Stage 1) *(ours)* | 28.08 |
| CLEL [26] | 8.61 | NCSNv2 [51] | 10.2 | HDEBM (Stage 2) *(ours)* | **21.82** |
| HDEBM (Stage 1) *(ours)* | 8.40 | QA-GAN [41] | 6.42 | | |
| HDEBM (Stage 2) *(ours)* | **8.06** | Diffusion Autoencoder [42] | 5.30 | Cond. EBM [11] | 43.7 |
| NCSNv2 [51] | 10.9 | COCO-GAN [28] | 4.0 | Diffusion + c.g. [8] | 30.46 |
| DDPM [21] | 3.2 | DDIM [50] | 3.5 | UHMC Diffusion + c.g. [8] | 26.89 |
| StyleGAN2-ADA [22] | 2.92 | PNDM [29] | **2.71** | Cond. BigGAN [3] | 6.02 |
| NCSN++ [52] | **2.20** | | | Cond. ADM + c.g. [7] | **2.97** |

a single step. As described in [48] the entire distillation takes about the same time as training the initial truncated diffusion. Since we are only training the truncated diffusion, compute is significantly less than required for full diffusion training. The diffusion network is frozen after training. We then train the first stage HDEBM using Algorithm 1. This is the most compute intensive part of training. Finally, we freeze the generator $G_1$ and initialize the energy network weights $\theta$ from the first stage weights to perform second stage training using Algorithm 2. This training converges rapidly and the cost is minor. See Appendix D for a thorough discussion of experimental details.

**Datasets.** We experiment with CIFAR-10 [24], Celeb-A [30] at resolution 64x64, and ImageNet [6] at resolution 128x128. Following standard procedure, we train and evaluate our models using only the training sets. For CIFAR-10 we trained the truncated diffusion using the first $T' = 256$ timesteps of a $T = 1000$ step diffusion with the cosine schedule from [7] and for Celeb-A and ImageNet we used the first $T' = 512$ timesteps of the same schedule. We use 4 A100 GPUs to train CIFAR-10 models and 8 A100 GPUs to train Celeb-A and ImageNet models.

### 4.1 Unconditional Generation

Our main experiments are unconditional generation on CIFAR-10, Celeb-A 64x64, and ImageNet 128x128. Table 1 presents FID scores for the first and second stages of our model, along with a comparison to a representative selection of existing models. Overall, our results show that HDEBM achieves state-of-the-art (SOTA) synthesis results among explicit EBMs for CIFAR-10 and Celeb-A. Furthermore, HDEBM achieves an FID score of 21.82 for unconditional ImageNet at 128x128 resolution which, to our knowledge, is SOTA for unconditional image generation without separate retrieval data.

To our knowledge, the generative modeling literature does not include a clear SOTA diffusion baseline for unconditional ImageNet at 128x128 resolution. At 256x256 resolution, unconditional ADM [7] achieves an FID score of 26.21 and RDM [2] achieves an FID of 12.21 with external retrieval data. RDM uses CLIP [43] encodings and therefore implicitly relies on the large-scale (text, image) dataset used to train CLIP. This complicates the unconditional modeling scenario. Modeling ImageNet at resolution 256x256 is beyond the 8 GPU budget used in this work and we hope to scale in future works for direct comparison to higher-resolution SOTA models.

For semi-unconditional ImageNet diffusion at 128x128 resolution, we include results from the recent work [8] which trains an unconditional diffusion model at 128x128 resolution and uses classifier guidance with standard reverse sampling (FID score of 30.46) and UHMC sampling (FID score of 26.89). It is likely that these models are not as highly optimized as ADM since FID scores at a lower resolution and with classifier guidance do not match unconditional ADM at a higher resolution. This highlights the difficulty of training highly optimized unconditional diffusion models. Overall, there is strong evidence that HDEBM can be competitive with or surpass highly optimized unconditional diffusion at 128x128 resolution. HDEBM will likely not match highly optimized retrieval augmented diffusion. We view the retrieval strategy as orthogonal to our approach and believe retrieval augmented HDEBM could yield further improvement in future work.

Table 2: Extended report of generative modeling metrics for ImageNet 128x128. Unconditional models are above the dividing line and conditional models are below the line. (*=re-evaluated using evaluation code from [7], c.g.=classifier guidance)

| | ImageNet (128 × 128) | | | | |
|---|---|---|---|---|---|
| Model | FID | sFID | IS | Precision | Recall |
| Hat EBM (small) [19]* | 43.89 | 9.63 | 21.21 | 0.43 | 0.44 |
| Hat EBM (large) [19]* | 31.89 | 7.39 | 26.03 | 0.54 | 0.45 |
| HDEBM (Stage 1) *(ours)* | 28.08 | 6.56 | 24.84 | 0.54 | 0.58 |
| HDEBM (Stage 2) *(ours)* | **21.82** | **5.08** | **28.56** | **0.56** | **0.59** |
| Cond. BigGAN [3] | 6.02 | 7.18 | 166.6 | **0.86** | 0.35 |
| Cond. ADM [7] | 5.91 | **5.09** | - | 0.70 | **0.65** |
| Cond. ADM + c.g. [7] | **2.97** | **5.09** | - | 0.78 | 0.59 |

Table 3: *Left:* Sampling speed and memory consumption of different models. Sampling speed is evaluated by the number of Equivalent NFE (ENFE) for full reverse sampling of popular diffusion networks. Memory utilization is assessed using an Nvidia A100 (80G) GPU. We set batch size to 32 for memory utilization experiments. *Middle and Right:* FID and sample compute comparison between HDEBM and accelerated diffusion methods for CIFAR-10 and CelebA.

| Dataset | Model | ENFE | Mem util |
|---|---|---|---|
| CIFAR-10 32x32 | DDPM++ | 1000 | 6.8G |
| | HDEBM-1 | 18.4 | 7.0G |
| | HDEBM-2 | 74.9 | 6.9G |
| CelebA 64x64 | ADM | 1000 | 18.6G |
| | HDEBM-1 | 9.8 | 31.8G |
| | HDEBM-2 | 114.5 | 29.4G |
| ImageNet 128x128 | ADM | 1000 | 55.3G |
| | HDEBM-1 | 6.3 | 48.2G |
| | HDEBM-2 | 41.2 | 45.7G |

| CIFAR-10 (32 × 32) | | |
|---|---|---|
| Model | ENFE | FID |
| DDIM [50] | 20 | 6.84 |
| ES-DDPM [32] | 100 | 5.52 |
| TDPM-GAN [60] | 1 | 7.34 |
| TDPM [60] | 5 | 3.51 |
| DPM-Solver(Type-3) [31] | 18 | 2.90 |
| Prog. Distill. [48] | 4 | 3.00 |
| Prog. Distill. [48] | 8 | 2.57 |
| HDEBM-1 *(ours)* | 18.4 | 8.40 |
| HDEBM-2 *(ours)* | 74.9 | 8.06 |

| CelebA (64 × 64) | | |
|---|---|---|
| Model | ENFE | FID |
| DDIM [21] | 10 | 13.73 |
| DDIM [21] | 100 | 6.53 |
| DPM-Solver(Type-2) [31] | 10 | 5.83 |
| DPM-Solver(Type-1) [31] | 20 | 2.82 |
| ES-DDPM [32] | 10 | 6.44 |
| ES-DDPM [32] | 100 | 3.01 |
| PNDM [29] | 10 | 7.71 |
| PNDM [29] | 100 | 2.81 |
| HDEBM-1 *(ours)* | 9.8 | 5.55 |
| HDEBM-2 *(ours)* | 114.5 | 4.13 |

Table 2 includes an extended report of generative modeling metrics to give a fuller picture of HDEBM performance on ImageNet 128x128. Notably, HDEBM outperforms conditional ADM and BigGAN in terms of sFID, and outperforms BigGAN in terms of recall. Overall HDEBM shows balanced and strong performance across evaluation metrics.

## 4.2 Sampling Compute Cost

Table 3 presents a comparative analysis of the Stage 1 and Stage 2 HDEBM with the standard diffusion model and accelerated diffusion methods, with respect to number of function evaluations (NFE), memory utilization, and FID on CIFAR-10 and Celeb-A. ImageNet is omitted due to lack of available unconditional models. Since sampling from our model involves both diffusion network forward pass (sometimes smaller than their typical size) and MCMC sampling, NFE is not directly applicable and we define a related metric ENFE (Equivalent NFE) that reports the sampling speed of our model in terms of the NFE of DDPM++ 32x32 and ADM 64x64 and 128x128 that take equivalent time. See Appendix D.2.1 for details. ENFE comparisons in Table 3 are rough comparisons because exact network structure varies between accelerated diffusions.

## 5 Conclusion

This work develops a hybrid of EBMs and diffusion models called HDEBM. A central component of the method is a truncated and distilled diffusion which can perform approximate MCMC on the data distribution using only a forward diffusion with negligible cost and a reverse DDIM process in a single forward pass. The truncated diffusion is incorporated between the generator and energy network of Hat EBM. The truncated diffusion can add fine details to the generator output for realistic synthesis while the MCMC trajectories learned by the EBM drive sample diversity to ensure good coverage of highly multimodal datasets. Experiments show that HDEBM yields high-quality image synthesis compared to explicit EBMs and SOTA FID results for unconditional ImageNet 128x128. HDEBM also has fast sampling speed compared to standard diffusion models. In future work we hope to further scale HDEBM to larger network sizes and higher resolution images, to investigate conditional and retrieval augmented models based on HDEBM, and to adapt HDEBM training to learn MCMC trajectories that have long-run stability and good mixing between modes.

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
