## A Broader Impacts

The primary motivation for the development of the HDEBM is to offer a state-of-the-art technique for the generation of highly multimodal distributions. Like all generative models, HDEBM holds potential for applications with complex implications. On the positive side, generative models, enable a new scope of creative and productive activity in various domains of work and expression. Online creators and influencers, individual creatives, and individual proprietors in business or e-commerce, will have access to very powerful creative tools that increase efficiency and reduce their costs. On the negative side, these models can also be harnessed for harmful purposes like the proliferation of misinformation and harassment activities. Generative models could lead to detrimental socioeconomic consequences such as displacement of human labor. Additionally, biases inherent in real-world datasets might cause trained generative models to contribute towards perpetuating discrimination and cultural exclusion. Moreover, while HDEBM has significantly improved sampling efficiency compared to standard diffusion models, its training and sampling process nonetheless demand substantial computational resources. For example, training HDEBM on ImageNet with dimensions 128x128 requires a minimum of eight A100 GPUs over several days. This requirement raises environmental concerns given the associated energy consumption and carbon emissions. These challenges are essential to address in the near future. Necessary steps include government regulation, responsible stewardship from corporations and individuals developing and applying generative models, and public education about generative models and their capabilities, limitations, and potential for misuse.

## B Limitations

A major limitation of HDEBM is the need to simulate short MCMC runs during each parameter update, which sometimes makes HDEBM learning slow compared to other types of models. This limitation is shared among a variety of MCMC-based EBM learning methods. HDEBM models for low-resolution datasets like CIFAR-10 take longer to train and do not match the quality of strong GAN or diffusion models. For higher resolution datasets like ImageNet, it becomes possible to balance the resources in our approach to achieve competitive training speed and superior generation quality with relatively small models. However, scaling HDEBM to larger model sizes poses a computational challenge.

Another consequence of training with short-run MCMC chains is the non-convergent learning outcome. The HDEBMs in our work should not be considered to be proper density models since evolving the MCMC trajectories for significantly more steps than used during training will severely degrade image quality. Instead, the model is essentially using the density objective to learn a generation process given fixed short-run sampling parameters and consistent initialization strategy. To our knowledge, this phenomenon is shared among all EBMs except for a few methods which intentionally focus on stabilizing long-run chains.

The current work only studies unconditional modeling since we expect that HDEBM will have the most potential for improvement over existing techniques in this area. Highly multi-modal unconditional modeling remains a challenging scenario which has received less attention compared to conditional modeling, and we hope our work can contribute to progress in this direction. Nonetheless, conditional modeling experiments with HDEBM are needed to gain a fuller picture of its capabilities.

## C Comparison with Related Methods

This section discusses similarities and differences between HDEBM and closely related generative models: Hat EBM [19] and truncated diffusion models for image generation [60, 32].

### C.1 Comparison with Hat EBM

HDEBM can be viewed as a version of Hat EBM that uses a specialized structure for the generator network. The components of the specialized generator are an initial generator network which is roughly analogous to the full generator of Hat EBM, followed by a second frozen generator component that adjusts the output of the first generator with a pseudo-MCMC update powered by a diffusion model. This choice of generator architecture is conceptually simple once understood but non-trivial to formulate. Integrating diffusion models into Hat EBM in a natural and effective way is the major

Table 4: Compute cost of HDEBM. For distillation, train steps represents the total number of steps across successive distillation stages. Compute is measured in A100 days.

| Dataset | Train Exp | Num GPU | Time per step (s) | Train Steps | Compute (A100 days) | Num Exps (approx.) | Total Compute (approx.) |
|---------|-----------|---------|-------------------|-------------|---------------------|--------------------|-------------------------|
| CIFAR-10 32x32 | Trunc Diff | 4 | 0.155 | 150K | 1.07 | 8 | 8.56 |
| | Distill | 4 | 0.195 | 160K | 1.44 | 6 | 8.64 |
| | Stage 1 | 4 | 0.81 | 250K | 9.38 | 45 | 422 |
| | Stage 2 | 4 | 1.86 | 20K | 1.72 | 8 | 13.8 |
| CelebA 64x64 | Trunc Diff | 8 | 0.215 | 250K | 4.98 | 4 | 19.92 |
| | Distill | 8 | 0.262 | 240K | 5.82 | 4 | 23.28 |
| | Stage 1 | 8 | 0.717 | 400K | 26.6 | 15 | 399 |
| | Stage 2 | 8 | 3.29 | 20K | 6.09 | 8 | 48.72 |
| ImageNet 128x128 | Trunc Diff | 8 | 0.210 | 400K | 7.81 | 10 | 78.1 |
| | Distill | 8 | 0.253 | 240K | 5.62 | 8 | 45.0 |
| | Stage 1 | 8 | 1.12 | 1500K | 155.6 | 35 | 5446 |
| | Stage 2 | 8 | 4.15 | 20K | 7.69 | 10 | 76.9 |
| Total | | | | | | | 6590 |

contribution of our work. HDEBM significantly outperforms Hat EBM across all metrics and datasets which shows the utility of our approach.

In terms of training compute, HDEBM imposes two further costs beyond those of Hat EBM: the cost of training the truncated and distilled diffusion model, and extra EBM training cost associated with forward and backward passes through both the base generator and diffusion model rather than the base generator only. See Table 4 for the cost of training the truncated and distilled diffusion. The time per batch of our Stage 1 training for CIFAR-10 and CelebA is approximately 20% slower than Hat EBM training due to the inclusion of the truncated diffusion. On the other hand, our HDEBM energy network for ImageNet uses the same structure as the small version of Hat EBM energy network so the time per batch of our ImageNet experiments is about twice as fast as training a large Hat EBM because the increase in cost from the diffusion model is offset by the reduced energy network size. The approximate MCMC from the truncated diffusion provides a high quality starting point that greatly eases the burden of the energy network when learning MCMC trajectories whose samples match the data.

## C.2 Comparison with Truncated Diffusion Generation Methods

This section compares HDEBM with the TDPM [60] and ES-DDPM [32] models which also use truncated diffusion to learn a generative model. The primary difference between our work and existing works is that we use truncated diffusion to train other components of our generative model (both the energy network and generator network $G_1$) while both prior works train the other components of their model separately from the truncated diffusion. In particular, TPDM trains a GAN to match the distribution $q_{T'}$ and then applies the truncated diffusion to obtain samples from $q_0$ and, ES-DDPM uses a variety of standard models trained on $q_0$ as base generators and performs a truncated $T'$-step forward and reverse diffusion process to refine samples. Since ES-DDPM uses both the forward and reverse process we interpret this as performing an approximate MCMC step on the samples of a base generative model to push samples closer to $q_0$. While the base generator in TDPM tries to match $q_{T'}$ and the base generator in ES-DDPM tries to match $q_0$, our base generator $G_1$ can match any distribution $q'$ such that performing a forward and reverse process on samples from $q'$ will approximately match sampling from $q_0$. We expect this task to be much easier than learning $q_0$ or $q_{T'}$ directly, especially when these distributions are highly multi-modal and options for the base generator are limited. Our work introduces the approach of learning additional networks that are adapted to a specific truncated diffusion network.

Distillation of truncated diffusion is another key difference between HDEBM and prior works. Distillation is essential for using truncated diffusion as a tool to teach other models because foward/backward passes through the truncated diffusion process are computationally infeasible unless the diffusion process uses only a few function evaluations. We distill all truncated diffusions to a single step. Experiments show that truncated diffusions can be distilled with significantly less loss of quality than distilled full diffusion. This is because truncated diffusions already have fewer steps to distill and because the denoiser predictions for samples at step $T'$ for the undistilled truncated diffusion already have many features of the original image which can be refined through distillation. In contrast,

denoiser predictions for an undistilled full diffusion at step $T$ have essentially no information about the original image and the distillation must teach the successive students the full DDIM image appearance for each initial noise sample.

The experimental settings in both prior works using truncated diffusion generally focus on cases where full diffusion and accelerated diffusions can achieve excellent results, such as CIFAR-10 and CelebA modeling. The conditional ImageNet experiments in both prior works obtain truncated diffusion models by truncating strong pretrained full diffusion baselines. In this work we focus on the highly multi-modal unconditional situation where full diffusion models face challenges and hope to show that truncated diffusion can help improve synthesis results beyond the capabilities of full diffusion. Truncated diffusion models are trained from scratch for each dataset, including ImageNet, to highlight the benefit of reduced learning complexity and compute cost required for truncated models compared to full models.

# D    Experimental Details

## D.1    Discussion of Compute

The approximate compute of each training experiment and the approximate number of times each experiment was performed is presented in Table 4. The numbers reported for a single experiment represent the compute cost of our final models presented in the experiment section. The runtime costs for the single experiments are precise. The total compute cost of each experiment is calculated by multiplying the cost of the final experiment by the approximate number of times each experiment was performed to get total compute estimates. This calculation is not precise due to differences between training runs and because many experiments were only partially completed before being terminated, often early in training. The total compute reports are very rough estimates which represent the order of magnitude of compute used.

Early development efforts focused on CIFAR-10 and we scaled up to CelebA and ImageNet once we obtained good results. We only trained the diffusion models and Stage 2 HDEBM models a few times and used most of our resources for Stage 1 HDEBM training. The most expensive experiment was Stage 1 HDEBM ImageNet training. A single full experiment of 1.5 million steps takes about 19 days on 8 GPUs, although good results can be obtained much earlier. We used approximately 64 A100 GPUs regularly over the course of 3 months to achieve our ImageNet results, including many partial experiments and debugging runs.

## D.2    Networks

The energy network $H$ uses the SN-GAN architecture [35] with ResNet blocks [17] with the modification that spectral normalization layers are removed. Attention is not used in the EBM because we did not observe noticeable improvement when included and because attention greatly slows sampling speed. The energy networks have the same structure as those used in Hat EBM. For CIFAR-10 modeling, we double the channel multiplier of the energy network. For ImageNet modeling, the energy network has the same structure as the small Hat EBM energy network, which is approximately one quarter the size of the large Hat EBM. Truncated diffusion networks $D$ use the DDPM++ UNet structure from [52] for CIFAR-10 at 32x32 resolution and the UNet structure from [7] for 64x64 resolution (CelebA experiments) and 128x128 resolution (ImageNet experiments). The ImageNet truncated diffusion uses a reduced channel width of 128 instead of 256 which is essential for efficient training speed and GPU memory usage. The generator network $G_1$ has the same structure as the truncated diffusion except for CIFAR-10 models, which use an SN-GAN generator with batch normalization removed as in Hat EBM.

### D.2.1    Details of Sampling Compute Comparison

To compare the sampling speed of HDEBM with various diffusion methods, we measure the speed of HDEBM image generation relative to the speed of diffusion model image generation using DDPM++ or ADM architectures depending on the resolution. This is done by generating 100 batches of samples with batch size 32 for both methods, where diffusion models use 1000 steps for generation. The average runtime of each method across 100 batches is recorded. Equivalent NFE (ENFE) for HDEBM is calculated by dividing the average HDEBM sampling time by the average diffusion model

Table 5: Hyperparameters of experiments. Experiments either used constant learning rates or cosine learning rates that decay to 0 over training. Experiments using cosine schedules also have an initial linear warmup phase starting from learning rate 0 over 10K steps. The reported learning rate indicates the fixed learning rate for constant schedules and the peak learning rate for cosine schedules. See Algorithms 1 and 2 for meanings of variables.

Diffusion Training and Distillation

| Dataset | $L_{\text{base}}$ | $L_{\text{distill}}$ | $B$ | $T'$ | LR type | $LR_{\text{base}}$ | $LR_{\text{distill}}$ |
|---|---|---|---|---|---|---|---|
| CIFAR-10 | 150K | 20K | 128 | 256 | constant | 1e-4 | 5e-5 |
| CelebA | 250K | 20K | 128 | 512 | constant | 1e-4 | 5e-5 |
| ImageNet | 400K | 20K | 128 | 512 | constant | 1e-4 | 5e-5 |

Stage 1 HDEBM

| Dataset | $L$ | $B$ | $K$ | $\epsilon$ | $N$ | $\beta_1$ | $\beta_2$ | LR type | $LR_{\text{energy}}$ | $LR_{\text{generator}}$ |
|---|---|---|---|---|---|---|---|---|---|---|
| CIFAR-10 | 250K | 128 | 50 | 5e-4 | 10K | 1 | 0.1 | constant | 5e-6 | 1e-4 |
| CelebA | 400K | 128 | 50 | 5e-4 | 10K | 1 | 0.1 | cosine | 1e-5 | 1e-4 |
| ImageNet | 1500K | 128 | 50 | 5e-4 | 10K | 1 | 0 | cosine | 1e-4 | 1e-4 |

Stage 2 HDEBM

| Dataset | $L$ | $B$ | $K$ | $\epsilon_1$ | $\epsilon_2$ | $\epsilon_3$ | LR type | LR |
|---|---|---|---|---|---|---|---|---|
| CIFAR-10 | 20K | 128 | 50 | 5e-4 | 5e-5 | 1e-3 | constant | 2e-5 |
| CelebA | 20K | 128 | 50 | 5e-4 | 5e-5 | 1e-3 | constant | 1e-5 |
| ImageNet | 20K | 128 | 50 | 5e-4 | 5e-5 | 1e-3 | constant | 1e-5 |

sampling time and multiplying by 1000. For other baselines in Table 3, we make the simplifying assumption that ENFE and NFE are equivalent for all diffusion models, which is not precisely true given variations in diffusion model architecture. Nonetheless, ENFE gives a reasonable estimate of the relative sampling speed of HDEBM, diffusion models, and accelerated variants.

### D.3   Training Details

This section describes training details for each type of experiment. Table 5 shows the hyperparameters used in each experiment. All experiments use Adam optimizers with default settings and also layer-wise gradient clipping by norm for network weight gradients with a max norm of 50.

Our diffusion models are straightforward. To truncate the model, we sample $t$ from a uniformly restricted discrete set $\{1, \ldots, T'\}$ for $T' < T$. All experiments use a truncated section of the discrete $T = 1000$ step cosine schedule from [7]. CIFAR-10 experiments set $T' = 256$, and CelebA and ImageNet experiments set $T' = 512$. All denoiser models predict the starting image rather than noise since previous work finds that this is more effective when the model is distilled. The diffusion loss weights are set as $w(\lambda_t) = e^{\lambda_t/2}$. The diffusion training implementation is intended to be minimal and no auxiliary loss terms or EMA is used. In practice, we did not find that highly optimizing the truncated diffusion model necessarily led to better results for later stages since the main performance bottleneck is Stage 1 HDEBM training. We leave further optimization of the truncated diffusion for future work.

### D.3.1   Stage 1 Training Details

Algorithm 1 gives pseudocode for Stage 1 HDEBM training. The Langevin update equation for the residual image $Y$ is

$$Y_{k+1} = Y_k - \frac{\epsilon^2}{2}\nabla_{Y_k} H(G(Z_1, Z_2) + Y_k) + \epsilon V_k \tag{12}$$

for $k = 1, \ldots, K$ and $V_k \sim N(0, I)$. Stage 1 ImageNet experiments used one further technique beyond what is described in the algorithm. We noticed that ImageNet training could sometimes become unstable due to a phenomenon where certain data images were assigned very low energy. This usually happened for images with stark features like bars and stripes. Instability can occur in this situation because the magnitude of the energy difference between data samples and MCMC samples affects the magnitude of the EBM weight update gradient. A few data states with excessively low energy can dominate the energy difference and cause spikes in EBM weight gradient magnitude which can destabilize training. Since we are working in the non-convergent learning regime and

not learning a proper density model, we expect this phenomenon is related to defects in the energy surface.

To avoid instabilities from this phenomenon we adopt a simple technique that filters out data samples with excessively low energy before calculating the loss gradient. This will prevent such states from being assigned even lower energy by the current update, and model forgetting should naturally cause their energy to eventually rise to the level of their peer states in which case they will once more be included in model updates. In practice, we calculate the median absolute deviation (MAD) for each batch of data energies and remove any data samples with energy lower than 6 MADs below the median data energy before loss gradient calculation. No filtering is performed on MCMC samples, and we only apply the technique for ImageNet models. This technique helps to train stable Stage 1 HDEBM models on ImageNet for 1.5 million steps and beyond.

---

**Algorithm 1** Stage 1 HDEBM Training

---

**Require:** Natural images $\{x_m^+\}_{m=1}^M$, EBM $U(x; \theta)$, generator $G(z_1, z_2; \phi) = G_2(G_1(z_1; \phi), z_2)$, Langevin noise $\epsilon$, number of shortrun steps $K$, generator loss weights $\beta_1, \beta_2$, initial weights $\theta_0$ and $\phi_0$, number of training iterations $L$, bank size $N$, batch size $B$.
**Ensure:** Learned weights $\theta_L$ for EBM and $\phi_L$ for generator.

Initialize bank of random latent states $\{Z_i^{(1)}, Z_i^{(2)}\}_{i=1}^N$ i.i.d. from the Gaussian $N(0, I)$.

Initialize image bank $\{X_i^-\}_{i=1}^N$ from generator using $X_i^- = G(Z_i^{(1)}, Z_i^{(2)}; \phi_0)$
**for** $1 \le \ell \le L$ **do**

    **Steps to Update EBM**
    Select batch $\{\tilde{X}_b^+\}_{b=1}^B$ from data samples $\{x_m^+\}_{m=1}^M$.

    Draw latent samples $\{\tilde{Z}_b^{(1)}, \tilde{Z}_b^{(2)}\}_{b=1}^B$ i.i.d. from the Gaussian $N(0, I)$.

    Initialize residual images $\{\tilde{Y}_{b,0}^-\}_{b=1}^B$ from the image with all pixels set to 0.

    Update residual images $\{\tilde{Y}_{b,0}^-\}_{b=1}^B$ with $K$ Langevin steps (12) obtain $\{\tilde{Y}_{b,K}^-\}_{b=1}^B$. Keep $\tilde{Z}_b^{(1)}, \tilde{Z}_b^{(2)}$ fixed.

    Sum generated image and residual image using $\tilde{X}_b^- = G(\tilde{Z}_b^{(1)}, \tilde{Z}_b^{(2)}; \phi_{\ell-1}) + \tilde{Y}_{b,K}^-$ to obtain negative samples $\{\tilde{X}_b^-\}_{b=1}^B$.

    Get learning gradient using (2) with samples $\{\tilde{X}_b^+\}_{b=1}^B$ and $\{\tilde{X}_b^-\}_{b=1}^B$ and update $\theta_{\ell-1}$ to get $\theta_\ell$.

    **Steps to Update Generator**
    Randomly choose unique indices $\{i_1, \ldots, i_B\} \subset \{1, \ldots, N\}$.
    Get paired batches $\{Z_{i_b}^{(1)}, Z_{i_b}^{(2)}\}_{b=1}^B$ and $\{X_{i_b}^-\}_{b=1}^B$ from $\{Z_i^{(1)}, Z_i^{(2)}\}_{i=1}^N$ and $\{X_i^-\}_{i=1}^N$.

    Get learning gradient using (10) with samples $\{Z_{i_b}^{(1)}, Z_{i_b}^{(2)}\}_{b=1}^B$ and $\{X_{i_b}^-\}_{b=1}^B$ and update $\phi_{\ell-1}$ to get $\phi_\ell$.

    Overwrite old states in bank with update $(Z_{i_b}^{(1)}, Z_{i_b}^{(2)}) \leftarrow (\tilde{Z}_b^{(1)}, \tilde{Z}_b^{(2)})$ and $X_{i_b}^- \leftarrow \tilde{X}_b^-$.
**end for**

---

### D.3.2 Stage 2 Training Details

Algorithm 2 gives pseudocode for Stage 1 HDEBM training. The Langevin update equations are

$$Y_{k+1} = Y_k - \frac{\epsilon_1^2}{2} \nabla_{Y_k} H(G(Z_{1,k}, Z_{2,k}) + Y_k) + \epsilon_1 V_{k,1} \tag{13}$$

$$Z_{1,k+1} = Y_{k+1} - \frac{\epsilon_2^2}{2} \nabla_{Z_{1,k}} H(G(Z_{1,k}, Z_{2,k}) + Y_{k+1}) + \epsilon_2 V_{k,2} \tag{14}$$

$$Z_{2,k+1} = Y_{k+1} - \frac{\epsilon_3^2}{2} \nabla_{Z_{2,k}} H(G(Z_{1,k}, Z_{2,k}) + Y_{k+1}) + \epsilon_3 V_{k,3} \tag{15}$$

for $k = 1, \ldots, K$ and $V_{k,1}, V_{k,2}, V_{k,3} \sim N(0, I)$. In practice, to save computation with the costly backward pass through $G$ we perform 5 steps of (13) for every one step of (14) and (15).

### D.4 Summary of Generative Metrics

FID [18] is our primary metric to measure a balance of sample quality and diversity. FID measures the Frechet distance between Gaussian approximations of the InceptionV3 embedding distribution of real and synthesized images. 50K model samples are used to calculate FID. 50K data samples are used when calculating FID for CIFAR-10 and Celeb-A and the full dataset statistics from the evaluation code provided by [7] are used for ImageNet. The ImageNet report includes additional metrics using the evaluation code from [7]: Inception Score [47], sFID [36], and precision and recall [25]. Inception Score measures the expected KL divergence between Inception classifier probabilities for synthesized images and the uniform prior label distribution. In practice, this metric is generally used as an indicator of quality but not diversity. sFID is similar to FID but uses the spatial layer embeddings rather than a fully connected layer embeddings of InceptionV3 to calculate Frechet

**Algorithm 2** Stage 2 HDEBM Training

**Require:** Natural images $\{x_m^+\}_{m=1}^M$, EBM $U(x;\theta)$, stage 1 pretrained generator $G(z_1, z_2) = G_2(G_1(z_1), z_2)$, Langevin noises $\varepsilon_1, \varepsilon_2$, $\varepsilon_3$, number of shortrun steps $K$, stage 1 pretrained weights $\theta_0$, number of training iterations $L$, batch size $B$.
**Ensure:** Learned weights $\theta_L$ for EBM and $\phi_T$ for generator.
  **for** $1 \le \ell \le L$ **do**
    Select batch $\{\tilde{X}_b^+\}_{b=1}^B$ from data samples $\{x_m^+\}_{m=1}^M$.
    Draw latent samples $\{\tilde{Z}_{b,0}^{(1)}, \tilde{Z}_{b,0}^{(2)}\}_{b=1}^B$ i.i.d. from the Gaussian $N(0, I)$.
    Initialize residual images $\{\tilde{Y}_{b,0}^-\}_{b=1}^B$ from the image with all pixels set to 0.
    Update initial residual images and latents with $K$ Langevin steps of (13), (14), and (15) to obtain $\{\tilde{Y}_{b,K}^-\}_{b=1}^B$, $\{\tilde{Z}_{b,K}^{(1)}, \tilde{Z}_{b,K}^{(2)}\}_{b=1}^B$.
    Sum generated image and residual image using $\tilde{X}_b^- = G(\tilde{Z}_{b,K}^{(1)}, \tilde{Z}_{b,K}^{(2)}) + \tilde{Y}_{b,K}^-$ to obtain negative samples $\{\tilde{X}_b^-\}_{b=1}^B$.
    Get learning gradient using (2) with samples $\{\tilde{X}_b^+\}_{b=1}^B$ and $\{\tilde{X}_b^-\}_{b=1}^B$ and update $\theta_{\ell-1}$ to get $\theta_\ell$.
  **end for**

Table 6: Variation in metric calculations for each experiment.

| | CIFAR-10 | CelebA | ImageNet | | | | |
|---|---|---|---|---|---|---|---|
| | FID | FID | FID | sFID | IS | Precision | Recall |
| Stage 1 | $8.43 \pm 0.02$ | $5.60 \pm 0.04$ | $28.20 \pm 0.12$ | $6.55 \pm 0.01$ | $24.95 \pm 0.08$ | $0.538 \pm 0.002$ | $0.574 \pm 0.005$ |
| Stage 2 | $8.12 \pm 0.04$ | $4.13 \pm 0.02$ | $21.91 \pm 0.11$ | $5.10 \pm 0.02$ | $28.53 \pm 0.02$ | $0.568 \pm 0.0004$ | $0.581 \pm 0.002$ |

distance. The precision and recall metrics approximate the data manifold and model manifold using overlapping hyperspheres whose radii are given by $k$ nearest neighbors of reference samples. The interior of the intersection of these hyperspheres defines the support of each manifold approximation. Precision and recall calculate the proportion of generated images that fall within the approximation of real image manifold and the proportion of real images that fall within the approximation of the model manifold, respectively.

### D.5   Variation for FID and Other Metrics

Since model generations vary between evaluations, FID and other metrics are subject to fluctuation. Many works in generative modeling will run FID evaluation several times and report the minimum value obtained as the final score. In this work, each metric was calculated three times and the main paper reports the results with the lowest FID score among these runs, along with extended metrics for ImageNet that are obtained with the lowest FID score. For a fuller perspective, Table 6 presents the mean and standard deviation of these three runs for each metric reported.

### D.6   Uncurated Model Samples

CIFAR-10 32x32

Stage 1        Stage 2

CelebA 64x64

Stage 1                                                 Stage 2

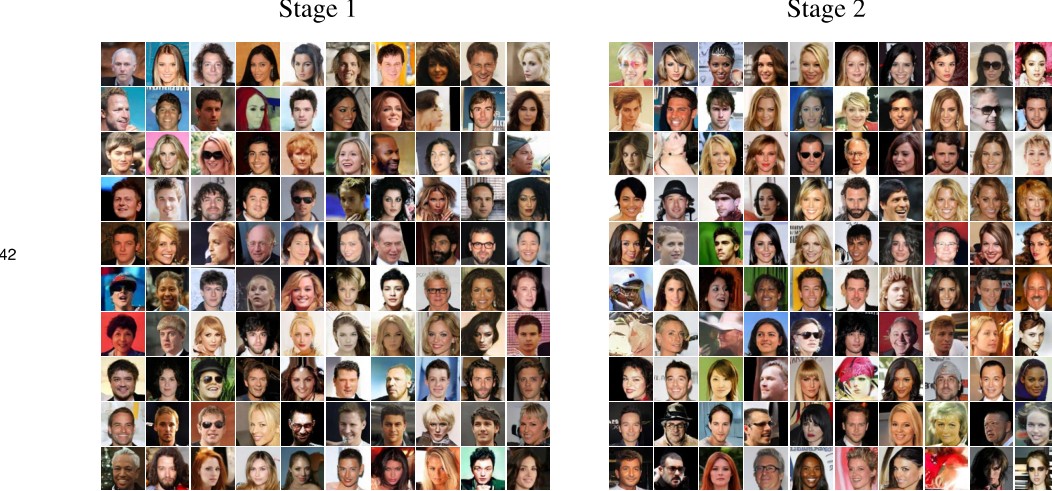

ImageNet 128x128

Stage 1                                                 Stage 2

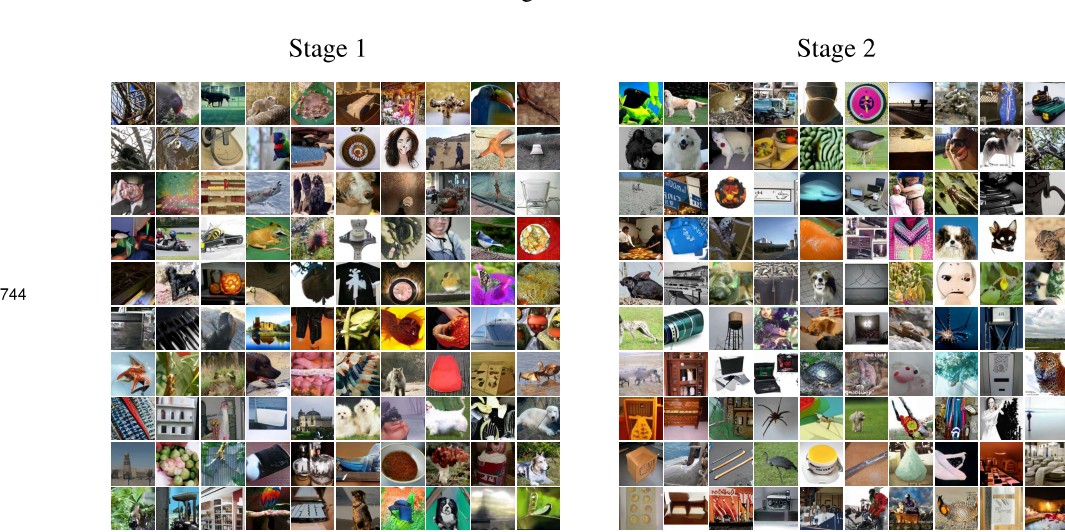