# OpenReview forum: "Tackling Unconditional Generation for Highly Multimodal Distributions with Hat Diffusion EBM"
_NeurIPS.cc/2023/Conference — Submitted to NeurIPS 2023_

### Official Review · Reviewer_JNTT · 2023-06-25

**Soundness:** 3 good
**Presentation:** 3 good
**Contribution:** 3 good
**Rating:** 6
**Confidence:** 3

**Summary:**

This work tries to improve the unconditional generation performance of Energy Based Model (EBM) by combining several techniques. It includes a pertrained diffsion model as a part of the generator and train the energy function and generator through cooperative learning. The performance of HDEBM outperforms many strong baselines of EBM.

**Strengths:**

The authors highlighted several strong techniques to improve the generative performance of EBM, including the introduction of pretrained diffusion model as a part of the generator, training the EBM under cooperative framework, sampling in both latent and noise spaces and 2-stage training. Some of these techniques are pre-exist and some are new. The final performance show good results among EBMs.

**Weaknesses:**

Currently, I have the following questions and concerns:

1. The proposed method appears to heavily depend on a well-pretrained (and distilled) diffusion model. While pretrained diffusion models exist for standard benchmarks like CelebA or ImageNet, pretraining and distilling such a model, especially on larger datasets or other specific applications, may pose greater challenges. This limitation could potentially restrict the application of the proposed method.

2. Additionally, how does the generative performance of the pretrained diffusion model used in this work compare? Does the Hierarchical Diffusion Energy-Based Model (HDEBM) achieve superior results compared to the pretrained diffusion model used for training?

**Questions:**

Here are my questions:
1. The impact of each step in the generative process: From my understanding, the image generation process involves three steps: the initial proposal of G1, the modification of G2 through the diffusion process, and the modification of EBM. Could the authors provide demonstrations of each generative step by showcasing a few samples and reporting the Fréchet Inception Distance (FID) for each step?

2. The understanding of pretrained diffusion model as MCMC sampling: Employing a diffusion model as MCMC sampling can alter the generative sample distribution to align it with the data distribution. However, as per the definition, negative samples from the Energy-Based Model (EBM) should originate from the EBM's own distribution. Although a well-trained EBM may generate samples from the data distribution, there can be a substantial gap between the EBM distribution and the data distribution during the training process. Hence, it begs the question of whether shifting the samples using a pretrained diffusion model could potentially introduce incorrect negative samples or shift the negative samples in an erroneous direction. Additionally, while the expression $D(\alpha x + \sigma z)$ holds true when $x$ follows the data distribution, it might not be valid when the initially generated $x$ deviates significantly from real data. How can we theoretically understand these two questions?

3. Missing reference:
It seems that the results of [1] is included in table 1 but the paper is not in the reference list;
Also authors may consider including the results of [2, 3] in table 1;

[1] A tale of two flows: Cooperative learning of langevin flow and normalizing flow toward energy-based model;

[2] Learning energy-based generative models via coarse-to-fine expanding and sampling.

[3] Learning Energy-Based Model with Variational Auto-Encoder as Amortized Sampler

**Limitations:**

The authors have adequately addressed the limitations of their work.

---

> ### Author Rebuttal · Authors · 2023-08-10
>
> Thank you for your thorough review; we have addressed your comments as follows:
> * *Limitations of training the diffusion model:* While training diffusion models on complex datasets can require significant computational resources, a major benefit of our method is that one only needs to learn a truncated diffusion which focuses on the parts of the diffusion trajectory which are easier to learn. Thus, our method could potentially be more amenable to scaling than standard diffusion models for increasingly complex data.
>
> * *Generation results compared to the pretrained diffusion:* We note that truncated  diffusions cannot be used to generate model samples directly since they require initialization from noisy data. Like previous works TDPM and ES-DDPM, we investigate ways in which a truncated diffusion can be incorporated into a larger generative model. In the global response, we replicate ADM on unconditional ImageNet at 128x128 and find HDEBM has superior performance compared to the standard diffusion model, and therefore also to accelerated variants.
>
> * *Investigation of difference phases of sampling:* Thank you for the very useful suggestion. We present FID scores and samples visualization for each phase of our Stage 1 ImageNet HDEBM in the global response.
>
> * *Possible mismatch between EBM and data samples:* This is certainly a valid concern. In general, it is difficult to robustly gauge the degree of separation from the data distribution that is still compatible with successful refinement by the truncated diffusion. Proposition 3.2.1 sheds some light on your question. Since a perfect truncated diffusion defines an MCMC trajectory with the data distribution as its steady-state, applying $G_2$ should still push samples towards the data distribution even when the initial samples differ from the data distribution (e.g. biased samples from imperfect EBM learning) . Applying $G_2$ to out-of-distribution states corresponds to the MCMC burn-in phase, while applying $G_2$ to in-distribution states corresponds to the MCMC steady-state phase. The empirical results show that HDEBM can eventually teach $G_1$ to generate samples which can be effectively refined by a truncated diffusion.
>
> * *Missing reference:* Thank you for pointing out that [1] is missing from the bibliography and for the suggestion to include [2,3]. We will make sure these works are properly cited in future revisions.

---

### Official Review · Reviewer_DvWX · 2023-07-04

**Soundness:** 2 fair
**Presentation:** 2 fair
**Contribution:** 2 fair
**Rating:** 4
**Confidence:** 4

**Summary:**

The paper proposes *Hat Diffusion Energy-Based Model (HDEBM)*, a hybrid model with a generator and an EBM component that can be primarily applied for unconditional image generation tasks. It is built upon the framework of Hat EBM, which produces the final image sample $X$ by combining (through addition) a raw image output from a generic generator network, $G(Z)$, with a further refinement step parameterized by a residual random variable, $Y$, which is obtained through Langevin sampling via an energy network, $H$; $i.e.$, $X=G(Z)+Y$. *HDEBM* applies an alternative parameterization of the generator component, $G$, by coupling the original Hat EBM generator $G_1$ with a truncated and distilled diffusion model $G_2$, thus produces the final image by $X=G_2(G_1(Z_1), Z_2)+Y$. The truncated and distilled diffusion model component $G_2$ is added to the original framework with the goal of driving the image output from $G_1$ closer to the true data distribution, thus can be viewed as an additional refinement step before the addition of $Y$. The paper demonstrates experiments results mainly in unconditional image generation on CIFAR-10, Celeb-A $64\times 64$, and ImageNet $128\times 128$, including *HDEBM* achieving a SOTA FID score of $21.82$ on the ImageNet $128\times 128$ dataset.

**Strengths:**

* The introduction and the related work section are well-organized in general, in terms of the clarity of the *HDEBM* framework overview and its connection with other works.
* The limitations of the work from a technical perspective and in terms of potential social impact are discussed in detail via Appendix A and Appendix B.
* The choice of diffusion-based generative models to improve the generator component of the Hat EBM framework for image generation tasks is a reasonable one, due to their strong performance in modeling ​​multimodal distributions.

**Weaknesses:**

* It’s hard to gauge the novelty of this work, since the whole *HDEBM* framework from objective functions to engineering details resembles Hat EBM closely. It essentially can be viewed as one specific parameterization of the Hat EBM framework, by substituting the original generator with a different one that includes a truncated and distilled diffusion model.
* The technical details of the diffusion model component are not very sound:
    * The term “forward/reverse” and “forward and reverse” have been used multiple times to describe the truncated diffusion process $G_2$; however, since the goal of $G_2$ is to refine the output of $G_1$, it’s expected to only run the reverse process of the diffusion model to “denoise” the image towards the true data distribution. Therefore, it’s not very clear on why the processes in both directions are run, especially since there is only one step after distillation.
    * The term “approximate MCMC step/sampling” has been used to describe the role of the truncated diffusion model, with a theoretical development via Proposition 3.2.1. However, the wording of the proof is quite concise and vague (directly jumps to stating “the process is aperiodic and irreducible” in Line 146 after assuming the diffusion model is “perfectly trained” in Line 145, as well as concluding $q_0$ being “a unique steady-state” in Line 150). It is already a well-established result that a diffusion probabilistic model is a Markov chain that aims to approximate the true data distribution, and the stationarity of such a chain shall be of less importance because we are not sampling as many timesteps as we want, but the same number during the reverse process as during the forward process (Ho et al., 2020). Therefore, the role of authors' "noting that an ideal truncated diffusion defines an approximate MCMC process with the data distribution as its steady-state" (Line 96-98) is confusing.
    * There are some factual errors in the authors’ claims about several related works:
        * In Line 152-153, the authors claim that “SDEdit [33] empirically observes that a truncated diffusion process can add realism to naive edits or rough user-defined templates.” However, truncated diffusion models were not mentioned in that work.
        * In Line 157-158, the authors claim that *DiffPure* “uses truncated diffusion to remove adversarial signals while preserving most of the original image appearance.” However, it appears that only conventional score-based diffusion models were used in that work.
        * In Line 302-303, the authors write, “As described in [48] the entire distillation takes about the same time as training the initial truncated diffusion.” But truncated diffusion was not mentioned in that work.
        * In Line 169-170, the authors write, “As noted in [48], a challenging aspect of learning a distilled diffusion is that the diffusion network output for a noise image at $t = T’$ provides essentially no information about the final state before distillation”, which appears to be counterintuitive since any intermediate diffusion step shall be viewed as a noisy version of the final state, but not a white noise. It’s not clear where in citation [48] that such statement was made.
* The design of combining two different generator networks appears to be a bit overcomplicated: there seems to be no obstacle in directly substituting the original generator with a model trained with the progressive distillation procedure, or consistency models (Song et al., 2023), if modeling multimodality is a desired property of this generator component under the Hat EBM framework.
* The content/structure of **EBM Learning** under Section 3.1 is very similar to Section 3.1 of (Hill et al., 2022), including the citations. Similarly, the background of diffusion models in Line 117-125 resembles Section 2 before Eq. (2) in (Salimans & Ho, 2022), and Line 233-234 resembles the text at the end of Sec. 3.2 of (Hill et al., 2022). This borders on text recycling and needs significant revision.


> Minor Issues
* Line 236 notation typo of $G(z)$: shall instead be “a fixed generator $G_2(x, z_2)$”?
* Repeated citation: [55] and [56].
* The pdf documents were uploaded as flat images, thus making them unsearchable and harder to read. It would be nice to upload a searchable and clickable pdf document in the future for reviewing.

[a] Yang Song, Prafulla Dhariwal, Mark Chen, and Ilya Sutskever. Consistency Models. In *Proceedings of the 40th International Conference on Machine Learning*, 2023.

[b] Mitch Hill, Erik Nijkamp, Jonathan Mitchell, Bo Pang, and Song-Chun Zhu. Learning Probabilistic Models from Generator Latent Spaces with Hat EBM. In *Proceedings of the 36th Conference on Neural Information Processing Systems*, 2022.

[c] Tim Salimans and Jonathan Ho. Progressive Distillation for Fast Sampling of Diffusion Models. In *Proceedings of the 10th International Conference on Learning Representations*, 2022.

[d] Jonathan Ho, Ajay Jain, and Pieter Abbeel. Denoising diffusion probabilistic models. In *Proceedings of the 34th Conference on Neural Information Processing Systems*, 2020.

**Questions:**

* Could the authors provide experimental results of unconditional generation with ImageNet $128\times 128$ by only a truncated and distilled diffusion model?
* Would the authors provide a reference for the challenge of diffusion models mentioned in Line 31-35?
* Why does the truncated and distilled diffusion model have both the forward and the reverse process, instead of just the reverse process that directly maps a noisy image to a sample closer to the data distribution?
* Eq. (7) the equation on the right: is $x$ a real data sample, or one generated by $G_1$?
* Can the authors provide a little more explanation on the sentence in Line 264-265, “The function of this loss term can be interpreted as training $G_1$ to invert $G_2$ given forward noise $Z_2$ and target image $X$.”? Specifically, what does “invert $G_2$” mean?
* In Stage 2 from Figure 2, there are multiple MCMC steps; why are the mapping from sample $Z_1$ and that from $Z_2$ marked as red arrows as well?
* In Line 289-290, the authors write, “we can perform MCMC on $z$ for the density (11) but not for (8)” – could the authors provide more explanations for it?
* How is the difference between generated images computed, $e.g.$, in Eq. (10) – are they the Euclidean distances in the original image space?

---
Update on 08/28/2023: I’ve increased my overall rating from 3 to 4:
* As the authors pointed out, the coordination of different moving parts of the framework is non-trivial, and their design choices to combine EBM models with diffusion models under the same framework could be learned from by other researchers. The framework achieves SOTA result on unconditional ImageNet at $128\times 128$ resolution, while Hat EBM did not, demonstrating the effectiveness of such design choices.
* The authors’ claim as their first main contribution that $G_2$ “defines an MCMC trajectory whose steady-state is the data distribution” is not very solid: in practice, it’s not clear on where the forward process of $G_2$ starts from ($X$ as the output of $G_1$ shall have a distribution quite different from $q_0$, otherwise there is no need to have $G_2$), or where the reverse process of $G_2$ ends up (shall no longer be $q_0$ even with a perfectly trained $D$). Reviewer iE1P expressed a similar concern as the second question, but I’m not convinced by the authors' response. Although the operation of adding noise then denoising by $G_2$ is an interesting design, it lacks clear motivation.

**Limitations:**

As mentioned before, the authors have sufficiently addressed the limitations from a technical perspective in Appendix B. Meanwhile, potential negative social impact has been discussed in Appendix A.

---

> ### Author Rebuttal · Authors · 2023-08-10
>
> We're grateful for the effort in reviewing our work and for your valuable suggestions. Our clarifications and responses follow below.
> * *Regarding novelty:* Like the related works TDPM and ES-DDPM, the primary novelty of our works comes from the design choices that we make to incorporate a truncated diffusion as part of a larger generative model. Our major technical novelty is using backpropagation through the truncated distilled diffusion both to train the generator network and to refine latent states to improve image quality in the Stage 2 HDEBM. Although HDEBM can be viewed as a special case of Hat EBM, integrating diffusion and EBM models in an effective way involves non-trivial design choices and insight into the learning process.
>
> * *Foward/Reverse used to describe $G_2$:* As shown in (7), the network $G_2$ will first add noise to the output from $G_1$ to create a noisy sample, and then denoise the noisy sample with $D$. Thus, $G_2$ performs both the forward and reverse process. This use of a truncated diffusion matches ES-DDPM but differs from TDPM, which predicts noisy images directly. We used the approach from ES-DDPM rather than the approach from TDPM because we found that the TDPM approach was ineffective when trying to model samples with large magnitudes of noise added, while the TDPM approach yielded results very similar to Hat EBM for small noise magnitudes. In our work, $G_1$ can learn to output any distribution such that samples from $G_1$ plus noise match noisy data samples. We find this is much easier than teaching $G_1$ to directly match true data (as in ES-DDPM) or to directly match noisy data (as in TDPM).
>
> * *Regarding Proposition 3.2.1:* Our claim about truncated diffusion is distinct from the observations in Ho et al., 2020. In particular, Proposition 3.2.1 describes the image space trajectory of a sample after repeated forward/reverse diffusion process are applied, which defines an MCMC trajectory whose steady state is the true data distribution under the assumption of perfect modeling. While diffusion updates for small timesteps $t$ of a standard diffusion model can also be viewed as approximate MCMC steps on the data distribution (since $q_t \approx q_0$ for small $t$), in practice the diffusion steps for small $t$ provide very minor change to the image. Applying a forward/reverse process of a truncated diffusion can yield much larger movement in the image space while still following the image manifold, as done in SDEdit and related works.
>
> * *Regarding factual errors:* We respectfully disagree that these are factual errors. SDEdit and DiffPure both use truncated forward/reverse processes which are directly analogous to our generator $G_2$, even though they do not use the term "truncated diffusion" explicitly. Although [48] distills full diffusion models, we find their observation about the cost of learning a distilled full diffusion also applies when learning truncated diffusion models (total cost of distillation stages is about the same cost as the base stage, whether the base stage is full or truncated). We will clarify the wording of line 302-303 in future revisions. Lines 169-170 refer to the paragraph on Page 5 of [48] which begins "Although this standard specification..." and describes the need to use alternate parameterizations for learning distilled diffusions.
>
> * *Why not just use Progressive Distillation?:* We find that our framework has superior performance compared to standard diffusion models on unconditional ImageNet 128x128 (see global response Table 1), in which case HDEBM would also outperform accelerated variants like Progressive Distillation.
>
> * *Similarities in Background Section:* We indeed adapt the background section notation and presentation from existing work to give a concise yet relatively complete description of previous works. In future versions, we will note that the background sections are generally based on the notations and presentation from Hat EBM and Progressive Distillation.
>
> * *Minor Issues:* In Line 236, $G(z)$ refers to the whole generator $G_2(G_1(z_1), z_2)$ where $z = (z_1, z_2)$. We will fix the repeated citation. We sincerely apologize for the unsearchable format, this occurred when we manually separated the main paper and appendix. Future revisions will be fully searchable.
>
> * *Generation with only truncated diffusion:* A truncated diffusion is not capable of serving as generative model in its own right because it requires initialization which match noisy data to generate realistic samples. Like TDPM and ES-DDPM, our work investigates ways of incorporating a truncated diffusion into a larger generative model.
>
> * *Challenges of unconditional diffusion modeling:* This refers to the gap in sample quality between unconditional and conditional diffusion models for high-resolution and highly multimodal data, which is widely known in the literature. Please refer to our response to a similar question from Reviewer HLKM.
>
> * *Eq. 7:* $x$ is a sample generated by $G_1$.
>
> * *Meaning of inverting $G_2$:* Minimizing the loss term $\| X - G(Z_1, Z_2 ; \phi) \|^2 = \| X - G_2(G_1(Z_1; \phi), Z_2) \|^2$ can be accomplished by tuning $\phi$ so that $G_2(G_1(Z_1; \phi), Z_2) \approx X$. Then one can view the update of $\phi$ as solving $G_1(Z_1; \phi) \approx X^{-}$ for some image $X^{-}$ such that $G_2 (X^{-}, Z_2) \approx X$.
>
> * *Red arrows in Stage 2 diagram:* These arrows indicate that we updated $Z_1$ and $Z_2$ using Langevin dynamics in Stage 2, in addition to the residual image $Y$.
>
> * *First Stage vs. Second Stage density:* Since $z$ appears in the normalization constant of (8), one cannot use standard MCMC techniques because this would require calculating the intractable normalizer for each update of $z$. The intractable normalizer of (11) does not depend on $z$ and one is free to use any MCMC technique which only requires the energy.
>
> * *Distance metric for (10):* Yes, this is Euclidean distance in the image space.

---

> > ### Comment · Reviewer_DvWX · 2023-08-16
> > **Response to Author Rebuttal**
> >
> > Dear authors,
> >
> > Thank you very much for your detailed response, it has helped me to better understand the paper and has clarified some of my questions. After incorporating other reviewers’ comments and rebuttal responses, I plan to keep my current score for now. I am open to make further adjustments during the second phase discussion period.

---

> > > ### Author Response · Authors · 2023-08-17
> > > **Thanks for your review**
> > >
> > > We greatly appreciate your thoughtful review and discussion. Please let us know if there are any points which we could address or improve upon which would assist with your assessment of our work.

---

### Official Review · Reviewer_HLKM · 2023-07-07

**Soundness:** 4 excellent
**Presentation:** 3 good
**Contribution:** 3 good
**Rating:** 6
**Confidence:** 2

**Summary:**

The authors propose the Hat Diffusion Energy-Based Model (HDEBM), which incorporates a distilled truncated diffusion model as a generator network for a Hat EBM. They note that a perfectly-trained truncated diffusion model can be used to define an MCMC process whose steady-state distribution is the data distribution. A two-stage training procedure is proposed. In the first stage, the energy network and generator networks are trained by performing MCMC on residual images conditioned on frozen latents. In the second stage, the energy network is finetuned so that both latents and residuals can be updated using MCMC. Empirically, HDEBM outperforms existing EBMs on ImageNet 128x128 and performs comparably to GANs and diffusion models on smaller resolutions.

**Strengths:**

- The work seems quite original. The idea of combining the strengths of diffusion models and EBMs for multimodal unconditional generation is a good one, and the procedure for unifying them into a single framework is nontrivial.
- Although proper comparisons are difficult to make, many experiments are run and results show HDEBM outperforms previous EBMs and achieves comparable results in image quality and sampling cost to GANs and some diffusion models.

**Weaknesses:**

- I found the presentation of the methodology section to be a bit confusing, and found myself referring back and forth between the main text and the appendix. It may be helpful if possible to bring several details from the appendix, such as the Stage 1 training algorithm, to the main text.
- The experiment section could be strengthened a bit. It may be compelling to compare with more recent diffusion models on CIFAR10 and CelebA, such as EDM [1]. Comparison with Hat EBM on CIFAR10 and CelebA would also be enlightening.

[1] Tero Karras, Miika Aittala, Timo Aila, and Samuli Laine. Elucidating the design space of diffusion-based generative models. In Advances in Neural Information Processing Systems, 2022

**Questions:**

- The authors mention that diffusion models have drawbacks when it comes to highly multi-modal unconditional modeling (lines 27-28). I may be misunderstanding this point, but this seems separate from the issue of there being few diffusion models to benchmark against at higher resolutions. What is the rationale for why multi-modal modeling is challenging for diffusion, and is there some basic experiment that could be included to demonstrate this?

- The proposition (Prop 3.2.1) regarding the connection between truncated diffusion and MCMC seems to be known and mentioned in previous works, as the authors mention, so characterizing this as a novel perspective may be too strong.
- The claim that HDEBM sampling costs are "significantly lower than diffusion models" (lines 13-14) seems unsubstantiated by the experimental results.

**Limitations:**

The authors adequately address the limitations of the work.

---

> ### Author Rebuttal · Authors · 2023-08-10
>
> Thank you for your thoughtful insights and suggestions to better our work. We have addressed each of your points below.
> * *Reorganization to improve clarity:* We agree that the clarity of our presentation could be improved by bringing details from the appendix into the main text. Future revisions of the text will focus on improving clarity and making sure that the main text is as self-contained as possible.
>
> * *Improving the experiment section:* We will include EDM results for CIFAR-10 and CelebA in Table 1 to provide additional context. Table 1 of the original submissions compares HDEBM and Hat EBM on CIFAR-10, CelebA, and ImageNet 128x128. Additional interpolation, reconstruction, and inpainting experiments can be found in the global response.
>
> * *Challenges of Diffusion for unconditional modeling:* Thank you for bringing up this point. We agree that the wording is somewhat unclear. For low-resolution data such as CIFAR-10, the gap in image quality between unconditional and conditional diffusion models is quite low. For high-resolution and highly multimodal data, there remains a large gap in quality between conditional and unconditional diffusion models. The gap between conditional and unconditional models is, in our view, evidence that unconditional modeling remains challenging for diffusion models. It is probably more clear to say that all current generative models, including diffusion models, face challenges for high resolution and highly multi-modal data. We will rephrase this section of the introduction to present this more clearly.
>
> * *Proposition 3.2.1 known in prior works?:* Although many prior works empirically observe that performing a partial forward and reverse diffusion process can add realism to an image, to our knowledge no work has explicitly observed that repeated forward/reverse processes of a perfectly trained diffusion defines an MCMC trajectory in the image space whose steady-state is the data distribution. We carefully checked SDEdit, DiffPure, TDPM, ES-DDPM, and others for such an observation but could not find it. Nonetheless, the diffusion literature is rapidly growing and it is difficult to ensure that we did not miss this observation in another work. If we are missing a reference which makes the same observation, we will gladly include it in future revisions and rephrase our contributions accordingly.
>
> * *Sampling costs compared to diffusion models:* Thank you for pointing this out. We meant that our method has significantly lower costs than standard (unaccelerated) diffusion models. In future revisions, we will rephrase this sentence to state the our work has comparable cost compared to accelerated diffusion models.

---

> > ### Comment · Reviewer_HLKM · 2023-08-19
> >
> > Thanks to the authors for the detailed responses. I feel my questions/concerns are addressed and I will retain my score of weak accept.

---

### Official Review · Reviewer_iE1P · 2023-07-07

**Soundness:** 3 good
**Presentation:** 3 good
**Contribution:** 3 good
**Rating:** 7
**Confidence:** 4

**Summary:**

Hat EBM introduced a framework to incorporate an arbitrary generator network $G : \mathcal{Z} \to \mathcal{X}$ (for example a GAN generator or a VAE) into an EBM by defining a joint energy function over the generator latent space and a residual image space that bridges between the generator output and the ground-truth data distribution. In particular, in HEBM an image is generated as $X = G(Z) + Y$ where $Z$ is a latent vector passed through a generator $G$, an $Y$ is a residual image. The joint energy function is $U(Y, Z ; \theta) = H(G(Z) + Y ; \theta)$ where $H(x; \theta)$ is a neural net that maps from images to scalars. Hat EBM models the joint distribution of the generator's latent space and the residual image, and can be used with either a pre-trained generator $G$, or can be used to learn $G$ in tandem with the energy function.

This paper introduces a variant of Hat EBM called Hat Diffusion EBM (HDEBM), that incorporates a diffusion model to partially denoise the output of the generator $G$. In particular, the authors augment HEBM with a diffusion model such that the generative process first samples a latent $Z_1$ and passes it through the generator function $G_1$ to produce an initial output image $G_1(Z_1)$. Then, as an additional step compared to HEBM, HDEBM adds noise $Z_2$ and runs one step of denoising using a pre-trained and distilled diffusion model, which yields $G_2(G_1(Z_1), Z_2)$. The application of the initial generator followed by a denoising step can be interpreted as a more complicated generator function $G(Z_1, Z_2)$. Finally, following Hat EBM, the final output image is formed by adding a residual, $X = G(Z_1, Z_2) + Y$.

For the diffusion component, the authors first train a truncated diffusion model from scratch, focusing on the less-noisy half of the denoising trajectory; they then use progressive distillation to distill the model into a single-step denoiser.

The authors propose a two-stage pipeline to train the HDEBM: the first stage assumes that $Z \sim \mathcal{N}(0, I)$ and learns the distribution of the residual images $Y$ conditioned on $Z$; the second stage learns the joint distribution of the generator latents $Z$ and the residuals $Y$. They show that the second stage leads to slightly improved results compared to only using the first stage.

Empirically, they use HDEBMs for unconditional image generation on CIFAR-10, CelebA, and ImageNet, and they compare FID scores to other classes of generative models (including other EBMs and diffusion models). They obtain competitive performance.

**Strengths:**

* Overall, the paper is well-written, clearly introducing the approach and discussing how it differs from Hat EBM.

* The proposed approach significantly outperforms Hat EBM in terms of FID scores.

* The idea is nice: HDEBM is an interesting way to incorporate ideas from diffusion models into an EBM, which could be useful inspiration for future work.

* Overall, the unconditional image generation experiments show very good performance, in particular compared to explicit EBMs. A good set of comparisons is provided, including other EBMs (Hat EBM, VAEBM, diffusion recovery likelihood EBM, VERA) and non-EBM methods like BigGAN.

**Weaknesses:**

* The paper does not clearly motivate the HDEBM approach. Why would one wish to use this complicated framework with several moving pieces (that need to be trained in stages) rather than simply using another class of generative model, in particular a diffusion model that also yields high sample quality without dropping modes?

* I think that the HDEBM approach should be further compared to the progressive distillation component that forms the denoiser $G_2$, in isolation. If Prog. Distillation is well-trained, then it has fast sample times and good FID scores, so what does HDEBM add to that?

* In Table 3, Prog. Distill significantly outperforms HDEBM both in terms of FID and sampling speed.

* The overall framework is fairly complicated, as it requires several components: the generator function $G_1$, the pre-trained, truncated and distilled diffusion model $G_2$, and the residual for the energy function. In particular, this leads to a complicated three-stage training pipeline, where first the diffusion model is trained and distilled, followed by the two stages of learning the energy function.

* The background in Section 3.1 should not be part of the method section, as it is not novel; this should be a new section between the Related Work and Method sections.

* In Eq. 3, it is strange to use $\epsilon$ to denote the step size and $V_k$ to denote the noise sample; usually $\epsilon$ would denote a noise sample, and the step size would be $\eta$ or $\alpha$.

* There seems to be inconsistent use of lowercase and capital $x$ to denote data samples (e.g., $x_t$ in Eq. 4 and $X_t$ in Eq. 5). Similarly, what is the difference between $z$ and $Z$ used in different parts of the paper (e.g., $G(z)$ in Eq. 8 and $G(Z)$ in the sentence just before Eq. 8)?

* I do not think that the discussion of Proposition 3.2.1 is clear. Among other things, the  proposition should clarify what is meant by a "perfect reverse process $D$", and clarify why the assumption that $D$ is perfect implies that there is always an image in $q_t$ that will map to a given image in the support of $q_0$.

* It feels like Section 3.2 is present to add math and formality to the paper, which does not seem necessary or helpful. This section does not inform the design decision of the HDEBM, it only serves as post-hoc justification for plugging in the diffusion model.

* I think that the paper should further clarify why two training stages are necessary rather than a single fused stage.

* Why are there no diffusion models listed in Table 2?

* Why is progressive distillation not reported in the CelebA part of Table 3?

* Can the diffusion model be fine-tuned during training of the HDEBM? If so, that would be an interesting ablation to see how much it helps.

* Figure 1 in this paper is almost identical to Figure 1 in Hat EBM; the only change is the addition of the "approximate MCMC" component $G_2$ and the corresponding random sample $Z_2$. I think this difference could be made clearer if the new components were highlighted, and the caption stated that the diagram was inspired by HEBM.

* The left side of Figure 2 is almost identical to Figure 2 in the Hat EBM paper; this should be mentioned in the caption, and the parts that are exactly the same or different should be highlighted somehow.

**Minor**

* L33 "network changes" --> "changes in the network weights (during training)"?

* L64 "curated samples" --> It would be nicer to see un-curated samples.

* Usually, $\mathcal{N}$ is used to denote a normal distribution rather than $N$. Also, typically $\mathbb{E}$ is the standard notation for expectation, not $E$ as used in the paper.

* L130 typo: "our works utilizes" --> "our work utilizes"

* L133 typo: "the full diffusion" --> "the full diffusion model"

* L161 typo: "values of $t$ greatly changes" --> "values of $t$ greatly change"

* Stages 1 and 2 should be more clearly delineated in Figure 2, for example with larger labels on top of the left- and right-hand sides.

* L266 typo: "match EBM" --> "match the EBM"

* In the caption of Figure 1: "passed through a forward/reverse truncated diffusion in $G_2$. $G_2$ then performs approximate MCMC on the data distribution."

* This wording is unclear: it sounds like first we perform forward/reverse diffusion and then the diffusion model performs approximate MCMC, but this is intended to say that the truncated diffusion is essentially doing approximate MCMC.

* L267: "We view this term as a regularizer." --> I think that this should be stated earlier, otherwise it sounds like both terms are equally important.



**Questions:**

* Is truncated diffusion used primarily to reduce the training time by considering only half the number of denoising steps? Have the authors investigated whether using non-truncated diffusion helps, at the cost of more compute?

* While empirically it seems to work, in principle wouldn't the output of $G_1$ potentially be out-of-distribution for the denoiser $G_2$? That is, $G_2$ is trained to denoise data that has been corrupted in a particular way (additive Gaussian noise), while the "corruptions" output by $G_1$ may be quite different. It might be interesting to see an ablation with respect to the amount of noise $Z_2$ added to $G_1(Z_1)$ before denoising. It seems useful to add $Z_2$ to bridge the OOD gap between the generated $G_1(Z_1)$ and the images $G_2$ was trained on.

**Limitations:**

* It is not clear why the paper focuses entirely on unconditional generation. Is it possible to extend HDEBM to the conditional setting? The original Hat EBM paper was applied in both conditional and unconditional settings.

* The experimental evaluation is somewhat limited, as only three main experiments are performed: unconditional image generation on CIFAR-10, CelebA, and ImageNet. The metrics focus on FID scores and compute/memory costs. But EBMs can do many other tasks, such as inpainting, OOD detection, etc. It might provide a more complete picture if the paper had more diverse experiments.

---

> ### Author Rebuttal · Authors · 2023-08-10
>
> We appreciate your thorough and positive review. Reponses to your main comments are below.
>
> * *Choice of EBM model family:* Please see our response to a similar question from Reviewer 88Mj.
>
> * *Comparison to the progressive distillation $G_2$ in isolation:* The truncated diffusion model $G_2$ can create realistic samples given noisy data samples, but it is not capable of serving as a generative model on its own. As in related works TDPM and ES-DDPM, the truncated diffusion in HDEBM must be a component of a larger generative model. Our works builds on TDPM and ES-DDPM to investigate ways in which a truncated diffusion model can be incorporated into a larger generative model.
>
> * *Better performance of Progressive Distillation:* Although we expected Progressive Distillation to outperform HDEBM on lower-resolution datasets, our experiments on higher-resolution unconditional datasets show that HDEBM can outperform standard diffusion models (and therefore accelerated variants like Progressive Distillation) in some scenarios.
>
> * *Complexity of the Pipeline:* We acknowledge that our pipeline is more complex than standard diffusion training. Our use of multi-stage training is consistent with current truncated diffusion works TDPM and ES-DDPM, which also train the final model in multiple stages.
>
> * *Background in Section 3.1 should not be part of the method section:* Thank you for pointing this out. We will move Section 3.1 into Section 2.
>
> * *Notation issues:* The confusion in notation in (3) comes from mixing notation common in the EBM literature with notation common in the diffusion literature. We will improve notation in future revisions. We aim to use lower case notation in cases where the variable plays the role of a constant and upper-case notation when the variable is a random variable. We will improve notation in revisions.
>
> * *Regarding Proposition 3.2.1:* Reasoning about the properties of a perfectly trained generative model can yield insights into the learning framework. For GAN models, the works [a, b] both reason about the properties of perfectly trained GANs and develop probabilistic interpretations (i.e. that ideal GANs minimize JSD and that composing the generator and discriminator of a perfect GAN defines an EBM). We view Proposition 3.2.1 in a similar vein. A perfectly trained diffusion model $D$ has the property that, if $X_t \sim q_t$, then $D(X_t, t) \sim q_0$ (see the "ODE Formulation" Section of EDM [c] for a similar discussion). Although truncated diffusions are used to add realism to initial images in several prior works (notably SDEdit), these works lack a theoretical perspective for why truncated diffusion increases realism. Our work provides an explanation for this phenomenon from the perspective of MCMC sampling, which is known to push out-of-distribution states towards the data distribution. We believe Proposition 3.2.1 adds important context for understanding truncated diffusions (in our own work and beyond) and could perhaps motivate development of MCMC samplers based on truncated diffusions.
>
> * *Reason for using 2 stages:* The first stage will draw an initial latent vectors from random noise, then add a residual image refinement while the latent vectors that define the generator output are fixed. The second stage will allow refinement of the latent vectors as well as the residual image. Performing MCMC in the latent space allows the model to find nearby latents which lead to significant better quality generated images $G(z)$.
>
> * *Diffusion Comparison in Table 2:* To our knowledge, there is no publicly available high-quality unconditional diffusion model trained on ImageNet at 128x128 resolution. To provide essential context for our results, we train an ADM model on unconditional ImageNet at 128x128 resolution. HDEBM generally has stronger performance.
>
> * *Similarity with Hat EBM Figures:* The similarity is intentional, as our work is closely related to Hat EBM. The suggestion to highlight the differences is very helpful and we will follow it in future revisions.
>
> * *Minor Issues:* Thank you for pointing out these issues, we will follow your suggestions. Uncurated samples for each model can be found in Appendix D.6.
>
> * *Motivation for use of Truncated Diffusion:* Yes, the primary reason for using a truncated diffusion is to alleviate the computation cost of the diffusion and shift resources towards initializing and refining the truncated diffusion. We expected that truncating a full diffusion could improve quality, as well as increasing the scale of the truncated diffusion.
>
> * *$G_1$ output might not match true corruptions:* This is a very valid concern and a critical part of our design. In our framework, $G_2$ will add noise to the output of $G_1$ before applying the denoiser $D$. Therefore the corruptions of the output of $G_1$ before denoising match ground-truth noise corruption.  Unlike TDPM, $G_1$ does not predict noisy images directly. Instead, $G_1$ can learn any distribution such that adding noise to the output of $G_1$ matches the distribution of noisy data. We experimented with directly predicting noisy samples with $G_1$ as in TDPM but found poor results for large noise magnitudes.
>
> * *Lack of Conditional Models:* In our experience, conditional EBMs often come with additional instability beyond their unconditional counterparts. We focus our efforts in a direction where we believe our method has the most potential compared with existing methods. Although Hat EBM uses the term "conditional Hat EBM" for one variant of the model, all experiments in Hat EBM are performed on unconditional data.
>
> * *Additional Experiments:* Additional interpolation, reconstruction, and inpainting experiments are included in Figure 1 of the global response.
>
> [a] Goodfellow et al., Generative Adversarial Nets.
> [b] Che et al., Your GAN is secretly an energy-based model..., 2020.
> [c] Karras et al., Elucidating the Design Space of Diffusion-Based Generative Models, 2022.

---

> > ### Comment · Reviewer_iE1P · 2023-08-15
> > **Response to Rebuttal**
> >
> > I have read the other reviews and the authors' rebuttal. I agree with other reviewers that the proposed method is fairly incremental compared to Hat EBM, but I think that HDEBM is a valid contribution that improves performance compared to other EBMs, however not necessarily compared to other diffusion models. I thank the authors for their responses, and for performing additional experiments in the rebuttal PDF. The authors have addressed most of my concerns.
> >
> > HDEBM is related to TDPM and ES-DDPM, and it would be good to add more discussion of these related works in the paper.
> >
> > I raised my score to 7.

---

> > > ### Author Response · Authors · 2023-08-16
> > > **Thanks for your time and guidance**
> > >
> > > Thanks again for taking the time and effort to provide a thorough and constructive review. We sincerely believe our paper will be significantly improved by incorporating feedback from yourself and other reviewers. We are very glad to hear that we have addressed most of your questions and that you have decided to raise your score. Future versions of the paper will provide more discussion of and comparison with the related works TDPM and ES-DDPM.

---

### Official Review · Reviewer_88Mj · 2023-07-30

**Soundness:** 2 fair
**Presentation:** 2 fair
**Contribution:** 2 fair
**Rating:** 4
**Confidence:** 3

**Summary:**

The authors describe a method to facilitate faster sampling in diffusion models whilst retaining quality, with a mix of energy and diffusion model. This consists of using an implicit generator, followed by noising then demonising from a retrained distilled diffusion model as a corrector, followed by an energy based model as an additional corrector. This is then trained in two stages, with the exception of the pre-trained distilled diffusion.

The authors introduce an MCMC approach whereby each step consists of adding noise and then removing noise using a demonising diffusion model.

**Strengths:**

The proposed methods obtains reasonably competitive performance across the tasks in question.

Taking gradients through truncated / distilled diffusions is interesting, and could be a useful contribution elsewhere.

Using noising then denoising diffusion model as a corrector is a nice insight and idea. Though similar approaches have already been considered in the literature as a way of correcting samples alongside the replacement conditioning method [1].

[1] RePaint: Inpainting Using Denoising Diffusion Probabilistic Models, Lugmayr  2022, https://openaccess.thecvf.com/content/CVPR2022/html/Lugmayr_RePaint_Inpainting_Using_Denoising_Diffusion_Probabilistic_Models_CVPR_2022_paper.html

**Weaknesses:**

If I have understood this correctly, the method appears quite incremental over existing truncated diffusion models [1]. The approach of [1] also trains an implicit generator for a noised version of the data, to which a diffusion is then applied as a corrector. Given the similarities I would appreciate a lot more discussion of how this approach differs and comparison experimentally in the main text. I understand some experiments are shown in Table 2 which suggests that [1] performs better in terms of FID than this approach, what is the reason for this given the similarities?

What is the benefit of the energy based parameterisation? The Langevin dynamics using the energy parameterisation is slow to sample as one needs to take a gradient at each step, this is at odds with the objective of accelerated sampling with distilled diffusion and truncation/ implicit generator, can the same Langevin dynamics not be applied using the score from the diffusion model at time T'? This is well known as "corrector" steps in the diffusion model literature [2].

Table 1 presents this work as an energy based model, it is not clear if that is really the case given the output is corrected with diffusion model. The distinction is not so clear to me. Table 1 shows the authors' method as top performing but neglects to show better performing EBM such as ones trained using score-matching in [3], which have similar performance to score-parameterised diffusion models.

Greater clarity is required around defining G, G_1, G_2, D. These are functions, networks etc. yet sentences such as "noising and demonising applied to G_2" then does not make much sense. The multi stage training, pre-training, and multi stage sampling is quite complicated and appears difficult to implement. It would be beneficial to include algorithms from the appendices in the main text to improve clarity.

Altogether clarity is an issue. Separating the training from the generation sections would be helpful.


[1] Truncated Diffusion Probabilistic Models and Diffusion-based Adversarial Auto-Encoders, Zheng et al 2022
[2] Score-Based Generative Modeling through Stochastic Differential Equations, Song et al 2021
[3] Should EBMs model the energy or the score?, Salimans et al 2021, https://openreview.net/forum?id=9AS-TF2jRNb

**Questions:**

See weaknesses above.

What is the benefit of this approach over truncated diffusion models or regular diffusion models? It appears sampling is roughly the same cost (including Langevin dynamics cost) and quality is questionable.

Perhaps this is an open review issue but the pdf text is not searchable, I do not have this issues on other papers I am reviewing. Is the text saved as an image in the pdf? This makes reviewing very difficult.

How is ENFE defined? Doesn't the MCMC steps essentially still involve neural function evaluations but also gradient so perhaps twice the cost of a single network evaluation?

Given the author's methods has been tried on imagenet128, other baselines should also be investigated on imagenet128 including ADM [7] given its better performance at higher resolution.

**Limitations:**

The authors include limitations and broader impact sections in the appendices, this probably ought to be in the main.

---

> ### Author Rebuttal · Authors · 2023-08-10
>
> Thank you for your time and your suggestions for improving our work. We agree that our central technical innovation is that taking gradients through the truncated and distilled diffusion, which allows us to learn energy and generator networks that are adapted to the truncated diffusion. We address your review comments and questions below.
>
> * *Relation with TDPM:* Our work is certainly related to TDPM and the similar work ES-DDPM. Unlike TDPM and like ES-DDPM, we add noise to generator samples before applying the truncated diffusion. The initial generator $G_1$ does not predict noisy images directly, but learns a distribution such that samples from $G_1$ plus additive noise match noisy data. We attempted to teach $G_1$ to directly predict noisy data but found this approach ineffective because it became difficult to directly model noisy data when large amounts of noise are added ($T'=512$ truncations steps as in our work), and our results were very similar to the base Hat EBM when small amounts of noise were added ($T'=100$ truncation steps as in TDPM). Unlike both TDPM and ES-DDPM, our auxiliary energy and generator are adapted to the truncated diffusion instead of learned independently, and we explore high-resolution generation with truncated models trained from scratch. Further comparison can be found in Appendix C.2. We will include more comparison with existing methods in the main paper. TDPM, as GAN-based method, outperforms HDEBM on small-scale datasets such as CIFAR-10 likely because GAN-based methods generally outperform EBM methods in this domain. Our central focus in this work is exploring high-resolution generation, where relative performance of GAN and EBM methods might not match low-resolution trends.
>
> * *Choice of EBM model:* We chose the Hat EBM parameterization for two reasons: 1) Hat EBM shows good performance relative to GANs with simple networks for high-resolution unconditional generation and 2) Hat EBM is compatible with a refinement stage (Stage 2 HDEBM), where the image appearance can be improved by movement in the latent space. We believe similar ideas could be applied to other kinds of generative models, but this is beyond our work's scope. EBMs are much smaller than the diffusion networks and an EBM Langevin step with a backward pass can take an order of magnitude less compute than a forward pass with a diffusion UNet. To keep compute low, our truncated distilled diffusions only require one forward pass to perform the entire the reverse process. More steps or predictor/corrector steps could yield improvement but we leave this for future work.
>
> * *EBM model family and related work [3]:* In this work, we use EBM to refer to a family of models that learns a single energy surface that represents the energy surface of the data distribution. Both stages of HDEBM incorporate all networks, including the truncated diffusion model, into a single unnormalized density in equations (8) and (11). Our work is derived from Cooperative Learning and Hat EBM, which both fall under the EBM umbrella. We view the work [3] as a diffusion model whose components are energy-based models. Although such distinctions are ultimately subjective, we believe [3] belongs more to the diffusion family than the EBM family, since the goal is to learn many models of noisy data at various noise levels instead of a single model of non-noisy data. We also note that recent EBM works such as CLEL do not include [3] among EBM baselines. We will include the reference [3] in future revisions as an "Energy-Based Diffusion" (in contrast to typical score-based diffusion).
>
> * *Definition of $G_1$, $G_2$, and $D$:* $G_1 (z)$ provides initial proposals from noise $z$. These samples, plus additive Gaussian noise, should match noisy data samples (line 206-207). $D(x)$ is a truncated distilled diffusion which can denoise a noisy data image $x \sim q_{T'}$ in a single forward pass (lines 188-190). $G_2 (x, z)$ is defined in (7), and it takes a sample $x$ from $G_1$, applies $T'$ steps of the forward process with noise $z$ to get a noisy sample, then denoises this noisy sample with $D$. "Noising and denoising through $G_2$" refers to the fact that $G_2$ contains both the forward process (noising) and the reverse process (denoising).
>
> * *Implementation complexity:* Our model is fairly straightforward to implement in practice. As in many EBM works, sampling during test-time is identical to sampling during training. We will shift implementation details from the appendix to the main paper.
>
> * *Benefit over diffusion models:* On one hand, our work seeks to extend high-quality generation to the EBM family. Learning (unnormalized) densities of high-dimension data is a long-standing open problem and our work is a step in this direction. On the other hand, our work shows that HDEBM can improve performance over diffusion models for highly multi-modal unconditional distributions.
>
> * *Text not searchable:* We sincerely apologize for this issue. It was caused by separating the appendix from the main paper manually. We will ensure that future versions are fully searchable.
>
> * *ENFE definition:* ENFE measures the runtime of HDEBM relative to a common fixed diffusion architecture. We generate samples from the HDEBM model and the reference diffusion model using 1000 timesteps for the diffusion model. Then ENFE is the ratio of HDEBM generation time over diffusion generation time, multiplied by 1000. HDEBM Langevin steps (backward pass included) can take a magnitude of order less compute than a forward pass with a large diffusion UNet.
>
> * *Comparison with ADM:* We agree that a comparison with ADM is essential for contextualizing our results. We were unable to complete a replication of ADM in time for the initial submission, but since that time has implemented an ADM baseline. Table 1 in the global response shows metric results for ADM on unconditional ImageNet at 128x128 resolution. HDEBM generally has stronger performance.

---

> > ### Comment · Reviewer_88Mj · 2023-08-17
> >
> > Thank you for your response.
> >
> > - **initial generator G1 does not predict noisy images directly, but learns a distribution such that samples from G1 plus additive noise match noisy data** \
> > This appears like a reparameterization of a generator targeting noising data i.e. the generator is the G1+additive noise.
> >
> > - **On the other hand, our work shows that HDEBM can improve performance over diffusion models for highly multi-modal unconditional distributions.** \
> > I am not sure where this work demonstrates this.
> >
> > - **EBMs are much smaller than the diffusion networks and an EBM Langevin step with a backward pass can take an order of magnitude less compute than a forward pass with a diffusion UNet.** \
> > This is a good point that should be emphasized, though the reason behind it is unclear to me.
> >
> > - **EBM model family and related work [3]:** \
> > Only one network is learned for the various energy models at each time t. Given an energy is learnt then it is an energy based model, in my opinion. This argument based on other works arbitrary distinction is not very strong in my opinion. A stronger argument for your method (and not [3]) would perhaps be in training time, or performance. Do these energy based diffusions not work as well as energy based models for energy tasks to more standard regular energy models?
> >
> > Overall it is still not clear to me what the benefit of this method is. If the objective is to learn an energy model for density estimation, then perhaps other tasks specific to energy models, such as density estimation tasks/ experiments should be shown rather than generative modelling and fid, where this approach appears more complicated (subjective / my opinion) and worse performing to diffusions.
> >
> > I do now believe there could be some benefit and interest in learning a high performance energy based model as opposed to diffusion, but I  think more work is needed in this paper to show this. I have increased my scores on soundness, but overall I believe more work is needed to justify a higher overall score.

---

> > > ### Author Response · Authors · 2023-08-17
> > > **Thanks for your review and continued discussion**
> > >
> > > Thanks again for your in-depth review and continued engagement in evaluating our work. Here are our responses to your further feedback.
> > >
> > > * *Reparameterization of the generator*: We agree that one interpretation of our method is to absorb the noise into $G_1$. In practice, we found this to be a crucial design choice compared to the TDPM approach where $G_1$ learns noisy data directly (without absorbing true Gaussian noise) and believe there is value in sharing this approach. Although ES-DDPM used a similar approach, all networks in that work are pretrained, while we explore ways to train auxiliary models that are fine-tuned to work in a coordinated way with a specific truncated diffusion.
> > >
> > > * *Improved performance of HDEBM over diffusion models*: Please see our global response, where we replicated ADM at 128x128 resolution on unconditional ImageNet and find that the smaller Stage 2 HDEBM and the larger Stage 1 and Stage 2 HDEBM outperform ADM in several metrics. Although a definitive comparison of generative models is very difficult, we believe that these results give very strong evidence that HDEBM significantly closes the low-resolution performance gap between EBM and diffusion models, and that HDEBM can outperform certain highly optimized implementations of diffusion models. In general, we believe that both HDEBM and ADM results can be improved by better architecture, better hyperparameters, more compute, etc. We believe our high-resolution results are valuable explorations in an area that has received much less attention than typical low-resolution benchmarks, especially in the EBM literature. To our knowledge, the larger Stage 2 HDEBM FID score of 17.03 on unconditional ImageNet 128x128 is SOTA in the literature. In future revisions, we will emphasize that definitive comparisons are extremely difficult and that our intention is not necessarily to definitively show improvement over diffusion models, but to greatly improve EBM learning to a point where it can become quite competitive with diffusion models at high resolutions.
> > >
> > > * *Smaller EBM size*: The EBM has a classifier-type structure similar to the encoder part of a UNet, without the expensive decoder and channel concatenation used in a typical UNet. Furthermore, we do not employ attention in the EBM for computational savings during MCMC sampling and instead rely on attention layers in the generator and truncated diffusion.
> > >
> > > * *Related work [3]*: While it is true that the model in [3] is a single network, this network parameterizes a family of energy functions $p_t (x; \theta)$ for different timesteps $t$, representing different distributions of noisy data. Even though these models are contained in a single network, we still feel that it is proper to describe them as a family of distinct EBM models. The models $p_t (x; \theta)$ for very small $t$ are indeed directly analogous to a "standard" single EBM model. However, since the learning for small $t$ uses Fisher Divergence with samples generated from small perturbations of the finite dataset, the energy surface for such models is learned only in a very small region around data samples. Although it is possible that information from larger $t$ could help develop the surface for smaller $t$, to our knowledge there is no theoretical justification to support this possibility. The use of negative MCMC samples in EBM learning allows HDEBM and related methods to develop an energy surface throughout the state space. Again, such a distinction is somewhat subjective, and we believe the "classification" of [3] will eventually be determined by common practice in the generative modeling community. Since [3] does not scale their method beyond CIFAR-10 and incurs significant computation cost beyond standard diffusion models from taking a second derivative of the score network during training which problematizes scaling, we do not feel that the classification of [3] either as an EBM or diffusion model significantly impacts the central claims or contributions of our work.
> > >
> > > We view learning a proper (unnormalized) density model as the long-term potential of EBM and we agree that HDEBM would be strengthened by more applications in this direction. Nonetheless, learning realistic synthesis is necessary condition of learning an accurate density and our work makes significant progress in this direction compared to other models in the EBM family. We believe our work is a valuable contribution with strong empirical results which would be of interest to the EBM and generative modeling community.

---

> > > > ### Author Response · Authors · 2023-08-17
> > > > **Global Response PDF might not be visible**
> > > >
> > > > It has just come to our attention that the PDF of our global author response might not be visible currently. We sincerely apologize for this, although we believe is it a technical error beyond our control since, as far as we know, the PDF seemed to be visible earlier in the review period. This global response contains the diffusion model comparison that we are referring to. We contacted the AC about this issue and will keep you and other reviews updated.

---

> > > > > ### Author Response · Authors · 2023-08-17
> > > > > **Global Response PDF visible again**
> > > > >
> > > > > It looks like the global response PDF is available once more. We apologize for any inconvenience.

---

> > > > ### Comment · Reviewer_88Mj · 2023-08-18
> > > >
> > > > The reason I mention [3] is that it appears the core argument of this work is that although your method is not competitive to diffusion models (e.g. diffusion model fid <~3, your method fig ~8 for cifar10), it is competitive for energy based models. However, [3] shows an energy based diffusion model with fid ~3 for cifar10. Note [4] scales the method of [3] to imagenet 128 (conditional) experiments. Though the method of [4] regarding composition may support this paper's argument for some justification of using energy based approaches over seemingly higher performance diffusions. I believe this point really needs to be made clearer.
> > > >
> > > > Please can you point out which table / figure supports *Improved performance of HDEBM over diffusion models*. Table 6 in the supplementary, from what I see, only shows your method.
> > > >
> > > > [4] Reduce, Reuse, Recycle: Compositional Generation with Energy-Based Diffusion Models and MCMC, https://arxiv.org/abs/2302.11552

---

> > > > > ### Author Response · Authors · 2023-08-18
> > > > >
> > > > > We agree that [3] has slightly better performance than our method on CIFAR-10, although from looking at the paper it seems that the FID score of [3] is 6.8 for the energy-based UNet, which is still some a distance from typical diffusion FID of <3 on CIFAR-10. We do not feel there is strong empirical evidence (yet) showing that energy-based diffusion can match typical diffusion models, although we agree this is an interesting and relevant direction, although somewhat tangential to our work. We emphasize that we already compared HDEBM to the strongest results from [4] in Table 1 of our main paper and find that our unconditional results for the smaller Stage 2 HDEBM significantly outperform the conditional results of [4] for both energy-based and standard diffusion models on ImageNet 128x128. Given that our unconditional results are stronger than the conditional results reported in [4], we feel that current evidence suggests that HDEBM is stronger than energy-based diffusion models for high-resolution unconditional generation. We believe this is an area that could use further exploration and acknowledge future results might paint a different story. Nonetheless, an in-depth exploration of energy-based diffusion is beyond the scope of this paper.
> > > > >
> > > > > The comparison we are referring to is a comparison between our re-implementation of ADM for unconditional ImageNet 128x128 (not performed by the original authors, but using their conditional ImageNet 128x128 settings and exact code) which is contained in Table 1 of our **author global response pdf**, which can be found by clicking on the "pdf" link in our standalone comment titled "Author Rebuttal by Authors" (please note that this year for NeurIPS, authors had the option to include a single-page PDF along with author responses). In this table, we compare against the smaller HDEBM in our main paper, as well as a larger HDEBM that we finished training after submission time. Stage 2 of our smaller HDEBM and both stages of the larger HDEBM outperform our reimplementation of ADM. We again stress that this is meant to show that HDEBM can be competitive with and in some cases outperform highly optimized implementations of diffusion models, although not necessarily as a definitive claim of improved performance. We also note that we included the diffusion results from [4] in Table 1 of the original paper and found our unconditional result for Stage 2 HDEBM outperformed their conditional results. These comparisons to ADM and [4] both provide evidence that HDEBM becomes competitive with (and sometimes might outperform) diffusion models for high-resolution highly unconditional distributions.
> > > > >
> > > > > Although diffusion models currently have the strongest performance among generative models, it is in the nature of research to explore alternative strategies even when established methods exist. We do not necessarily feel that HDEBM has to have stronger performance than diffusion models to make useful contributions to the study of EBM. There are many papers similar to ours which study EBMs primarily from the perspective of synthesis, and our work makes progress in this established area while providing novel ways to integrate diffusion models and EBMs, along with novel insights into how to understand and apply truncated diffusion for generative modeling.

---

> > > > > > ### Comment · Reviewer_88Mj · 2023-08-18
> > > > > >
> > > > > > I am very sorry for my oversight, I was looking at the wrong file. And thank you for pointing to the reference on [4] in your main paper (it is difficult to navigate as not searchable) and the correction regarding FID in [3]. I recalled similar energy/ diffusion FID scores, but I assumed the authors of [3] had competitive diffusion FID score around ~3, not both energy / diffusion ~6.
> > > > > >
> > > > > > Thank you for the detailed discussion it has really helped me better understand the paper. The reason I probe is that I am trying to ascertain the selling points and hence benefit to the neurips community. The methodological contribution over truncated diffusion and hat ebm is not so significant (in my opinion) hence probing for performance motivation over EBM / diffusion.
> > > > > >
> > > > > > To summarise:
> > > > > > This work introduces an energy based model consisting of a reparameterized truncated diffusion + energy model, whereby the diffusion component is pretrained and distilled.
> > > > > >
> > > > > > The generative procedure consists of
> > > > > > 1) an implicit generator
> > > > > > 2) pretrained distilled diffusion corrector
> > > > > > 3) energy based model Langevin dynamics
> > > > > >
> > > > > > The training is
> > > > > > 0) pretrain and distill a diffusion
> > > > > > 1) Stage 1 to train an implicit generator by reconstruction
> > > > > > 2) Stage 2 to train the energy based model.
> > > > > >
> > > > > > Methodological contributions are:
> > > > > > - taking gradients through a distilled diffusion is feasible to train the generator
> > > > > > - reparameterizing the truncated diffusion implicit generator
> > > > > > - adding an energy based model to the truncated diffusion and training as an energy based model
> > > > > >
> > > > > > Other strengths:
> > > > > > - Empirical performance shows SOTA for unconditional Imagenet-128 amongst both diffusion and energy approaches
> > > > > > - There may be benefits of having an energy as opposed to a diffusion
> > > > > >
> > > > > > Weaknesses:
> > > > > > - lack of clarity in the paper
> > > > > > - unclear motivation, regarding the benefit of energy based in particular over diffusion (empirical results aside)
> > > > > > - complex training and sampling procedure
> > > > > > - limited methodological novelty (subjective)
> > > > > > - open question over inconsistent cifar10 performance, given excellent imagenet128 performance
> > > > > >
> > > > > >
> > > > > > I have increased my score to borderline reject. I recognise the benefit of the approach and appreciate the author's additional experiments, it certainly strengthens the paper, but I believe the paper is not yet ready for publication.

---

> > > > > > > ### Author Response · Authors · 2023-08-18
> > > > > > > **Thanks so much for your thorough review and discussion.**
> > > > > > >
> > > > > > > We truly appreciate the effort and level of depth of your review and discussion. Communicating with yourself and other reviewers have led to very constructive suggestions for improving the presentation and evaluation of our work. The summary in your recent comment is a fair and accurate assessment of our work. Improving the clarity and motivation of our work will be the central focus as we prepare the revision text. The strong ImageNet performance relative to CIFAR-10 performance is somewhat surprising, although in our view not necessarily a weakness, but rather an interesting kernel for further study which builds upon a similar observation by Hat EBM. While we respect the reviewer's view that our work is not ready for publication, we believe that it is ready for publication.

---

### Author Rebuttal · Authors · 2023-08-10

Thanks to all reviewers for their time and insightful comments and suggestions. Our paper will certainly benefit from incorporating reviewer feedback in future revisions. Our global response includes:

* a larger scale HDEBM experiment
* results of ADM for unconditional ImageNet 128x128
* additional reconstruction, image completion, and interpolation experiments
* further investigation of the samples from different steps of our model

---

### Decision · Program_Chairs · 2023-09-21

**Decision:**

Reject

**Comment:**

The paper proposes to combine the diffusion model and the Hat EBM for generative modeling. Strong empirical results are shown to demonstrate the effectiveness of the model. The paper has obtained mixed ratings after the rebuttal and discussion. One reviewer (Reviewer HLKM) assigned a score of Weak Accept with notably low confidence, leading AC to assign low weight to this particular rating and comment. Reviewer iE1P found that the idea was interesting and results were good, thus leaning to accept. Reviewer 88Mj recommended rejecting the paper because of unclear motivation and limited innovation. Reviewer DvWX also recommended rejecting the paper because of the incremental innovation and writing issue. Some concern regarding a proof of the paper is not fully addressed by the author in the rebuttal. Reviewer HLKM and Reviewer JNTT, who assign a score of borderline accept, also found that the major concerns raised by Reviewer 88Mj and Reviewer DvWX were valid and reasonable. During an internal discussion among reviewers and AC, four reviewers have a consensus on the major concerns that the paper’s contribution is incremental and some technical and writing issues are unresolved. The AC carefully reviewed the paper, its rebuttal, and discussions.  The AC found that the arguments of Reviewer 88Mj and DvWX are persuasive. The AC believes that the paper is not yet ready for publication and recommends a rejection in its current state. The AC encourages the authors to enhance their paper for future submissions, with the goal of presenting a more compelling submission in the next venue.